# Phospholipid flippase ATP11C is endocytosed and downregulated following Ca$^{2+}$-mediated protein kinase C activation

Hiroyuki Takatsu[1], Masahiro Takayama[1], Tomoki Naito[1], Naoto Takada[1], Kazuya Tsumagari[2], Yasushi Ishihama [2], Kazuhisa Nakayama [1] & Hye-Won Shin [1]

We and others showed that ATP11A and ATP11C, members of the P4-ATPase family, translocate phosphatidylserine (PS) and phosphatidylethanolamine from the exoplasmic to the cytoplasmic leaflets at the plasma membrane. PS exposure on the outer leaflet of the plasma membrane in activated platelets, erythrocytes, and apoptotic cells was proposed to require the inhibition of PS-flippases, as well as activation of scramblases. Although ATP11A and ATP11C are cleaved by caspases in apoptotic cells, it remains unclear how PS-flippase activity is regulated in non-apoptotic cells. Here we report that the PS-flippase ATP11C, but not ATP11A, is sequestered from the plasma membrane via clathrin-mediated endocytosis upon Ca$^{2+}$-mediated PKC activation. Importantly, we show that a characteristic di-leucine motif (SVRPLL) in the C-terminal cytoplasmic region of ATP11C becomes functional upon PKC activation. Moreover endocytosis of ATP11C is induced by Ca$^{2+}$-signaling via Gq-coupled receptors. Our data provide the first evidence for signal-dependent regulation of mammalian P4-ATPase.

[1] Department of Physiological Chemistry, Graduate School of Pharmaceutical Sciences, Kyoto University, Sakyo-ku, Kyoto 606-8501, Japan. [2] Molecular and Cellular BioAnalysis, Graduate School of Pharmaceutical Sciences, Kyoto University, Sakyo-ku, Kyoto 606-8501, Japan. Correspondence and requests for materials should be addressed to H.-W.S. (email: shin@pharm.kyoto-u.ac.jp)

Lipid bilayers of cellular membranes exhibit asymmetric lipid distributions. In mammalian cells, the aminophospholipids phosphatidylserine (PS) and phosphatidylethanolamine (PE) are abundant in the cytoplasmic leaflet, whereas phosphatidylcholine (PC) and sphingomyelin (SM) are enriched in the exoplasmic leaflet of the plasma membrane[1–3]. Phospholipids are mostly synthesized on the cytosolic side of the endoplasmic reticulum (ER) and newly synthesized lipids must be scrambled across the bilayer to the luminal leaflet to avoid the imbalance of phospholipid mass[4], although scrambling proteins in the ER have not yet been identified. Phospholipids are distributed throughout organelle membranes and the plasma membrane, and thus newly synthesized phospholipids are transported to other organelles via phospholipid transfer proteins, or via vesicular transport. PS is synthesized on a region of the ER, called MAM (mitochondria-associated membranes), and converted to PE in mitochondria[5]. PS in the cytosolic leaflet of the ER could be transported to the cytosolic leaflet of the plasma membrane by exchange of PS with phosphatidylinositol 4-phosphate at the ER-plasma membrane contacts[6]. PS is also found in the luminal side in earlier secretory compartments, although PS is mostly distributed in the cytosolic leaflet in late secretory compartments such as the *trans*-Golgi network, late endosomes and the plasma membrane[7]. Since type IV p-type ATPases (P4-ATPases) translocate aminophospholipids from the exoplasmic/luminal to the cytosolic leaflets of cellular membranes[8,9], the presence of P4-ATPases in these organelles[10] is consistent with the asymmetric distribution of phospholipids in these membranes. PS is flipped to the cytosolic leaflet at the *trans*-Golgi network by P4-ATPases and the PS-flipping is required for the secretory vesicular transport[11,12]. In addition, PS is abundant in the cytoplasmic side of the plasma membrane and recycling endosomes[7,13,14], and plays important roles in the recruitment and/or activation of regulatory proteins, such as protein kinase C (PKC), K-Ras, Cdc42, Rac1, and EHD1, for signaling, cell polarity, cell migration, and membrane trafficking[14–18].

In previous studies, we showed that the human P4-ATPases ATP11A and ATP11C localize to the plasma membrane and flip NBD-labeled PS (NBD-PS) and NBD-PE, whereas ATP8B1, ATP8B2, and ATP10A flip NBD-PC specifically at the plasma membrane[9,19,20]. We also showed that those P4-ATPases interact with CDC50A, which is required for their transport from the ER to the plasma membrane in HeLa cells[10,20]. ATP11A and ATP11C are expressed ubiquitously in human and mouse[21]. ATP11C is a major PS-flippase in certain cell types such as CHO-K1 and KBM-7 cells, leukocytes, and erythrocytes[19,22–24]. ATP11C deficiency causes a defect in B-cell maturation, altered erythrocyte shape, anemia, and hyperbilirubinemia[25–27].

Regulated exposure of PS in the exoplasmic leaflet is critical for several biological processes, including apoptotic cell death, platelet coagulation, fusion of muscle cells, and activation of lymphocytes[28–33]. PS exposure during aggregation of platelets is triggered by a $Ca^{2+}$-dependent scramblase, TMEM16F, and its mutation gives rise to Scott syndrome[34,35]. $Ca^{2+}$-regulated exocytosis in neuroendocrine chromaffin cells, PC12 cells, and neurons is accompanied by disruption of phospholipid

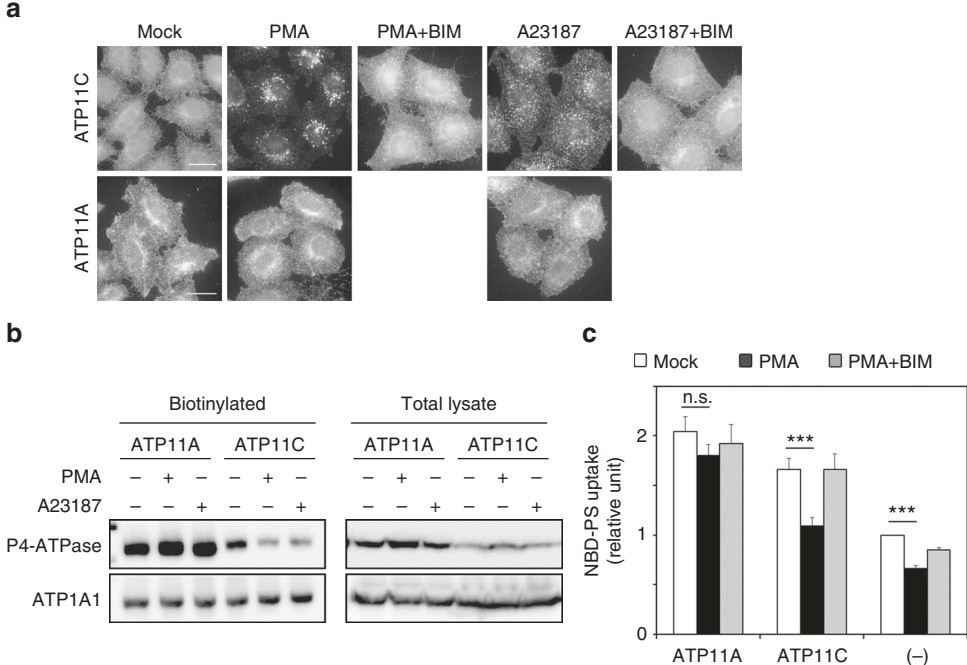

**Fig. 1** ATP11C is endocytosed by treatment with phorbol 12-myristate 13-acetate (PMA) and increasing cytosolic $Ca^{2+}$. **a** HeLa cells stably expressing C-terminally HA-tagged ATP11A, and ATP11C were treated for 15 min at 37 °C with vehicle alone (Mock); with either 400 nM of PMA (PMA) or PMA and 2 µM of BIM-1 (PMA + BIM); or with 1 µM A23187 in the presence of either 1.8 mM of $CaCl_2$ (A23187) or 1.8 mM $CaCl_2$ and 2 µM BIM-1 (A23187 + BIM). The cells were fixed and immunostained with anti-HA antibody, followed by Cy3-conjugated anti-rat secondary antibody. See Supplementary Fig. 1 and Supplementary Movie 1. **b** Cell-surface expression levels of ATP11A and ATP11C following treatment with PMA or A23187 and $CaCl_2$ were analyzed after surface biotinylation. Proteins precipitated with streptavidin-agarose beads were subjected to immunoblot analysis (left panels, biotinylated). 15% of the input of the biotinylation reaction was loaded in each lane (right panels, total lysate). Expression of ATP11A and ATP11C proteins was analyzed by immunoblotting with anti-HA and anti-ATP1A1 (as an internal control) antibodies. **c** HeLa cells stably expressing HA-tagged ATP11A and ATP11C, and parental cells (−) were treated with vehicle alone (white bars), 400 nM of PMA (black bars), or 400 nM of PMA and 2 µM of BIM-1, simultaneously (gray bars) for 15 min. The cells were then washed with flippase assay buffer and incubated with NBD-PS at 15 °C for 5 min. After extraction with fatty acid-free BSA, the residual fluorescence intensity associated with the cells was determined by flow cytometry. Fold increase of NBD-PS uptake is shown relative to that in Mock-treated parental HeLa cells (−). Graph displays averages from four independent experiments ± SD. ***$p < 0.001$

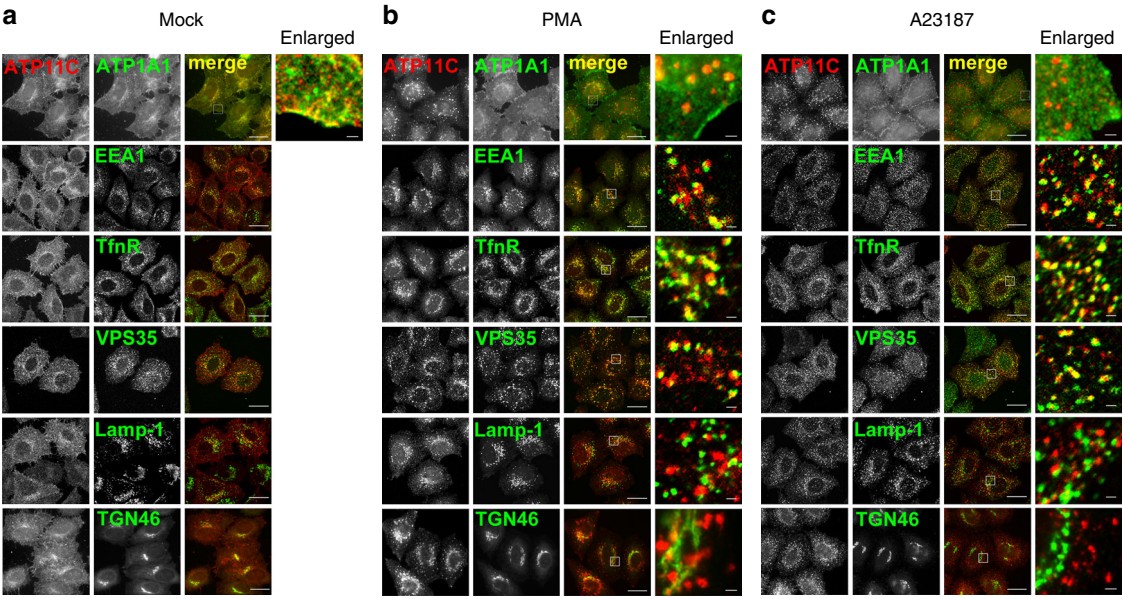

**Fig. 2** Internalized ATP11C is localized to early/recycling endosomes. **a–c** HeLa cells stably expressing C-terminally HA-tagged ATP11C was treated for 15 min with vehicle alone (**a**), 400 nM PMA (**b**), or 1 μM A23187 in the presence of 1.8 mM of $CaCl_2$ (**c**). The cells were fixed and doubly stained for HA and organelle markers: ATP1A1 (for the plasma membrane); EEA1 (for early endosomes); transferrin receptor (TfnR); VPS35, a component of retromer complex (for early/recycling endosomes); Lamp-1 (for late endosomes/lysosomes); and TGN46 (for *trans*-Golgi network). The cells were then incubated with Cy3-conjugated anti-rat and Alexa Fluor 488-conjugated anti-mouse, anti-rabbit, or anti-goat secondary antibodies. Images (except top and bottom panels) were obtained by confocal microscopy. Scale bars, 20 μm. Insets are enlarged (bars, 1 μm)

asymmetry, resulting in the externalization of PS in the outer leaflet of the plasma membrane[36]. In apoptotic leukocytes, PS exposure is promoted by the activation of Xkr8, as well as the inhibition of the PS-flippase ATP11C[23,37–39]. The increase in cytosolic $Ca^{2+}$ level in human erythrocytes inhibits incorporation of aminophospholipids[40], and $Ca^{2+}$-dependent PKCα activation mediates PS exposure along with scramblase activation and flippase inhibition[41,42]. Therefore, regulated exposure of PS might be accomplished by inhibition of PS-flippases as well as activation of scramblases, but it remains unclear how the PS-flippase activity is regulated spatiotemporally in response to specific signals in living cells, but not in cells fated for disposal such as activated platelets, red blood cells, or apoptotic cells.

Here we show that ATP11C is endocytosed following treatment of cells with phorbol ester or an increase in cytosolic $Ca^{2+}$ level, in HeLa and Ba/F3 cells. ATP11C is also endocytosed following treatment of cells with serotonin or histamine probably through $Ca^{2+}$ signaling via Gq-coupled serotonin or histamine receptor. Moreover, we reveal a characteristic motif for endocytosis, SVRPLL, which acts as a di-leucine motif ([DE]XXXL[LI])[43,44] upon PKC activation. We further demonstrate that the signal-responsive endocytosis of ATP11C is important for regulation of its PS-flippase activity at the plasma membrane.

## Results

**ATP11C is internalized following PMA treatment or increasing cytosolic $Ca^{2+}$.** Spatiotemporal PS exposure at the plasma membrane occurs not only in cells fated for disposal, such as apoptotic cells and activated platelets, but also in many activated cells, such as lymphocytes, neuroendocrine chromaffin cells, and neurons[31–33,36,45]. Therefore, there must be a signal-dependent and spatiotemporal regulation of externalization and internalization of PS at the plasma membrane. Unexpectedly, we discovered that ATP11C, a PS-flippase and localizing to the plasma membrane[9] (Fig. 1a and Supplementary Fig. 1, Mock), was delocalized from the plasma membrane to intracellular

endosomal structures by treatment of HeLa cells stably expressing HA-tagged ATP11C with phorbol 12-myristate 13-acetate (PMA), a potent PKC activator (Fig. 1a). Moreover, ATP11C also localized to intracellular structures following an increase in the cytosolic $Ca^{2+}$ level by treatment of cells with a $Ca^{2+}$ ionophore A23187 in the presence of extracellular $CaCl_2$, although to a lesser extent (Fig. 1a). We then investigated whether ATP11A was delocalized by PMA treatment, because, like ATP11C, it also localizes to the plasma membrane where it specifically flips PS and PE[9,10]. In marked contrast to the relocalization of ATP11C, the localization of ATP11A to the plasma membrane was not altered by PMA or A23187 treatment (Fig. 1a). We also investigated the dynamic localization of other P4-ATPases that localize to the plasma membrane. As shown in Supplementary Fig. 1, none of them, except for ATP11C, delocalized from the plasma membrane by PMA or A23187 treatment. Importantly, the translocation of ATP11C from the plasma membrane to endosomal structures induced by PMA and A23187 was abrogated by simultaneous treatment with bisindolylmaleimide (BIM)-1, a PKC-specific inhibitor, indicating that the translocation of ATP11C was mediated by $Ca^{2+}$-dependent PKC activation (Fig. 1a). We confirmed the level of cell-surface expression of ATP11C and ATP11A by surface biotinylation analysis. Upon PMA treatment, the level of ATP11C on the cell surface decreased, whereas the ATP11A level slightly increased (Fig. 1b). We then visualized the endocytosis of EGFP-tagged ATP11C in HeLa cells by time-lapse imaging analysis (Supplementary Movie 1). EGFP-ATP11C was localized to the plasma membrane at steady state, but upon treatment with PMA rapidly internalized and accumulated to endosomal compartments. Together, these findings show that ATP11C is endocytosed in response to $Ca^{2+}$-dependent PKC activation.

**Flippase activity toward PS decreased by PMA in HeLa cells.** We next investigated whether the PS-flippase activity in the plasma membrane is affected by treatment of cells with PMA.

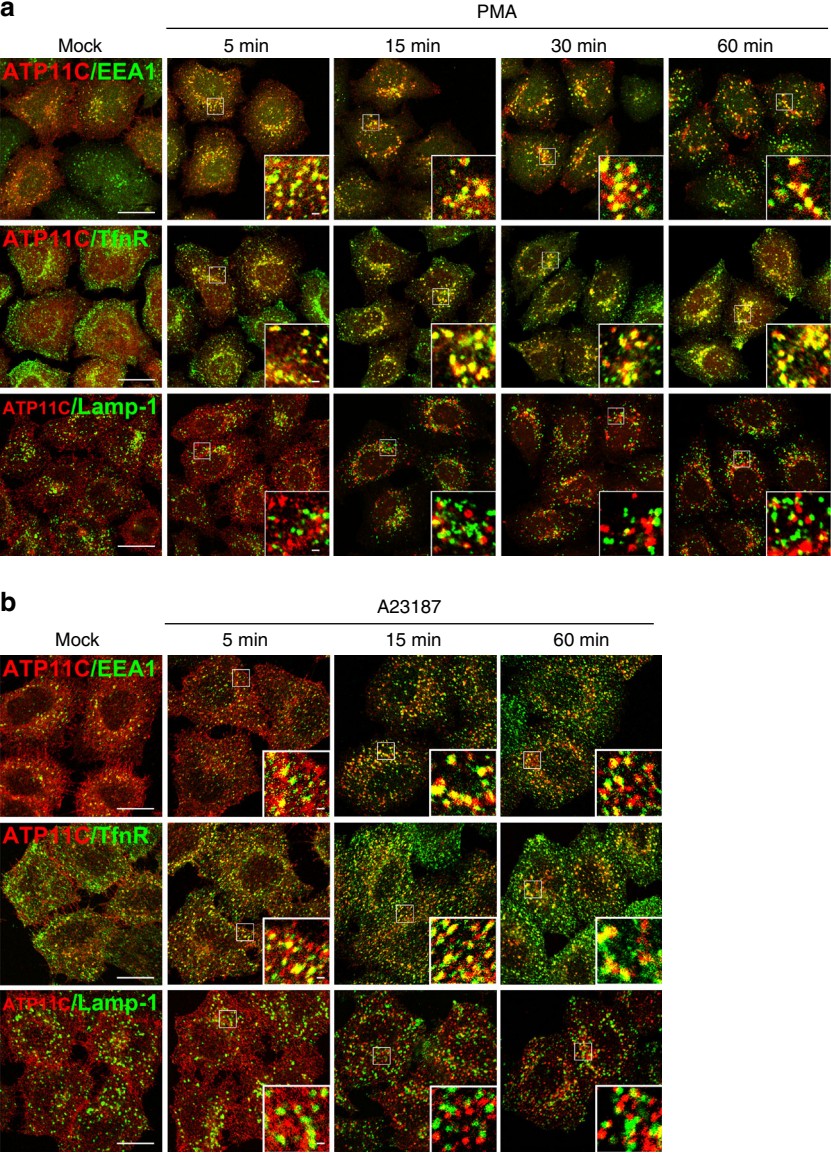

**Fig. 3** Internalized ATP11C is not transported to late endosomes/lysosomes. **a**, **b** HeLa cells stably expressing C-terminally HA-tagged ATP11C was treated with vehicle alone for 60 min (Mock), 400 nM PMA ((**a**) PMA), or 1 μM A23187 in the presence of 1.8 mM CaCl₂ ((**b**) A23187) for indicated times. The cells were fixed and doubly stained for HA, and EEA1, TfnR, or Lamp-1, followed by incubation with Cy3-conjugated anti-rat and Alexa Fluor 488-conjugated anti-mouse secondary antibodies. Images were obtained by confocal microscopy. Scale bars, 20 μm. Insets; bars, 1 μm

Incorporation of NBD-PS increased in HeLa cells stably expressing ATP11A and ATP11C, in comparison with the parental cells (−)[9] (Fig. 1c, white bars). Importantly, the PS-flippase activity in cells expressing ATP11C, but not ATP11A, decreased upon PMA treatment (Fig. 1c, black bars). Moreover, the reduced NBD-PS flipping activity was restored by simultaneous treatment of ATP11C-expressing cells with PMA and BIM-1 (gray bars), indicating that the inhibition of the PS-flippase activity was dependent on PKC activation. Because ATP11C, but not ATP11A, was endocytosed upon PMA treatment (Fig. 1a, b), the decrease in the PS-flippase activity is attributable to sequestration of ATP11C from the plasma membrane. Notably, the basal PS-flippase activity in parental HeLa cells (−) also decreased upon PMA treatment, potentially due to downregulation of endogenous ATP11C.

**Internalized ATP11C localizes to early/recycling endosomes.**
To identify the endosomes where ATP11C localizes following

PKC activation, we performed immunostaining for ATP11C along with several organelle markers such as EEA1 for early endosomes, transferrin receptor (TfnR), and Vps35 for early/recycling endosomes, Lamp-1 for late endosomes/lysosomes, TGN46 for the *trans*-Golgi network (TGN), and ATP1A1 for the plasma membrane. PMA treatment increases early endosomal size[46]; accordingly, EEA1-, TfnR-, and Vps35-positive structures slightly increased in size upon treatment of cells with PMA (Figs. 1a and 2b). ATP11C was extensively colocalized with ATP1A1 in control cells (Fig. 2a), but hardly colocalized with ATP1A1 in PMA- or A23187-treated cells (Fig. 2b and c). Following PMA treatment, internalized ATP11C was extensively colocalized with EEA1, TfnR, and Vps35 (Fig. 2b, enlarged insets), but minimally colocalized with Lamp-1 (Fig. 2b, enlarged inset). An increase in cytosolic Ca²⁺ by A23187 treatment also caused translocation of ATP11C from the plasma membrane to early/recycling endosomes (Fig. 2c, enlarged insets), but not to late endosomes (Fig. 2c, enlarged inset). Although the size of

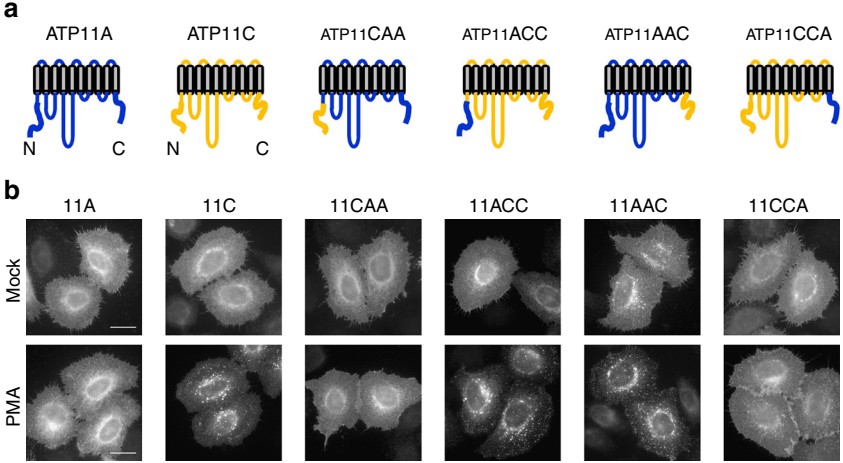

**Fig. 4** C-terminus of ATP11C is indispensable for PMA-induced endocytosis. **a** Schematic structures of a chimeric construct of ATP11A and ATP11C proteins are shown. **b** HeLa cells were transiently co-transfected with expression vectors for FLAG-tagged CDC50A and HA-tagged ATP11A, ATP11C, ATP11CAA, ATP11ACC, ATP11AAC, or ATP11CCA. Cells were treated with vehicle alone (Mock) or 400 nM of PMA (PMA) for 15 min and fixed. The fixed cells were permeabilized and incubated with anti-HA antibody, followed by Cy3-conjugated anti-rat secondary antibody. Scale bars, 20 μm

endosomes did not increase, in contrast to their behavior in PMA-treated cells, the number of small endosomes in cell periphery increased in A23187-treated cells. Internalized ATP11C was not colocalized with TGN46. Therefore, PKC activation caused translocation of ATP11C to early/recycling endosomes, but not to late endosomes.

Next, we examined internalization of ATP11C over a time course (Fig. 3). ATP11C was localized to EEA1-positive early endosomes at early time points (5 min) following PMA or A23187 treatment, suggesting that ATP11C was readily endocytosed upon PKC activation (Fig. 3 and Supplementary Movie 1). Moreover, internalized ATP11C was not delivered to late endosomes, but remained in early/recycling endosomes even after 60 min (insets), suggesting that internalized ATP11C was not transported to late endosomes/lysosomes for degradation, but was accumulated at early/recycling endosomes and could probably be recycled to the plasma membrane (see also Fig. 10d).

**C-terminal cytoplasmic region of ATP11C is indispensable for its PMA-induced endocytosis.** P-type ATPases, including yeast P4-ATPases, possess a regulatory domain in their N- or C-terminal cytoplasmic region[10,47–51]. We therefore asked whether the N- or C-terminal cytoplasmic region of ATP11C is responsible for the PMA-induced endocytosis. To address this question, we constructed chimeric ATP11A and ATP11C (Fig. 4a and Supplementary Fig. 2). In the ATP11CAA and ATP11ACC constructs, the N-terminal cytoplasmic regions of ATP11C and ATP11A were exchanged, whereas in the ATP11AAC and ATP11CCA constructs, the C-terminal cytoplasmic regions were exchanged. We transiently expressed these chimeras in HeLa cells, and observed their localization following PMA treatment. In vehicle-treated cells, the chimeras localized to the plasma membrane as efficiently as wild-type (WT) ATP11A and ATP11C (Fig. 4b, Mock). After PMA treatment, ATP11CAA remained localized to the plasma membrane, whereas ATP11ACC was endocytosed and localized to endosomes (Fig. 4b). Thus, the N-terminally exchanged chimeras behaved similarly to WT ATP11A and ATP11C (Fig. 4b) upon PMA treatment. In striking contrast, C-terminally exchanged chimeric mutants behaved in an opposite manner: upon PMA treatment, ATP11AAC was endocytosed and localized to endosomes, whereas ATP11CCA was not able to internalize and remained at the plasma membrane (Fig. 4b).

These results revealed that the C-terminal cytoplasmic region of ATP11C is responsible for the PMA-induced endocytosis.

We next asked whether endocytosis of ATP11C is responsible for the decrease in its PS-flippase activity at the plasma membrane and the endocytosis is also observed in other cells. To this end, we made use of Ba/F3 cells stably expressing ATP11A, ATP11C, and the ATP11AAC mutant (Fig. 5). Consistent with the data obtained with HeLa cells, ATP11C and ATP11AAC were endocytosed upon treatment of Ba/F3 cells with PMA or A23187 in the presence of extracellular $CaCl_2$ but ATP11A was not (Fig. 5b, c). We then measured flippase activities toward NBD-PS, -PE, and -PC after treating cells with PMA or both PMA and BIM-1. As shown in Fig. 5d–f, cells expressing ATP11A, ATP11C, and ATP11AAC exhibited elevated flippase activities toward NBD-PS and NBD-PE, but not NBD-PC, in comparison with parental Ba/F$_3$ cells (−) (white bars). PS/PE-flippase activities in ATP11C-expressing cells decreased dramatically upon treatment of cells with PMA in comparison with cells treated with vehicle only (Fig. 5d, e). Importantly, PS/PE-flippase activities in ATP11AAC-expressing cells also dramatically decreased after PMA treatment, whereas those of ATP11A-expressing cells did not change (Fig. 5d, e). Moreover, the reduced flippase activities of cells expressing ATP11C and ATP11AAC were restored by simultaneous treatment with PMA and BIM-1. These results strongly suggest that the C-terminal region of ATP11C is critical for PKC-induced endocytosis, and that the decreases in PS/PE-flippase activities can be attributed to endocytosis of ATP11C and ATP11AAC by PKC activation. We failed to examine flippase activities affected by an increase in cytosolic $Ca^{2+}$; because $Ca^{2+}$-dependent scramblase is activated upon treatment of cells with A23187 in the presence of extracellular $CaCl_2$[34], NBD-phospholipids uptake displayed by flippases was not accurately measured. Flippase activities toward PS and PE, but not PC, decreased upon treatment of parental Ba/F3 cells with PMA (Fig. 5d–f, (−)), suggesting that endocytosis of endogenous ATP11C might be responsible for the decrease in PS/PE-flipping activity. We previously showed that ATP11C prefers PS to PE[9]; accordingly, the PMA-induced inhibition of endogenous flippase activity toward PE is milder than that toward PS (Fig. 5d, e, (−)). A higher flippase activity toward PE was exhibited in cells expressing ATP11AAC and ATP11A, as compared with that in ATP11C-expressing cells

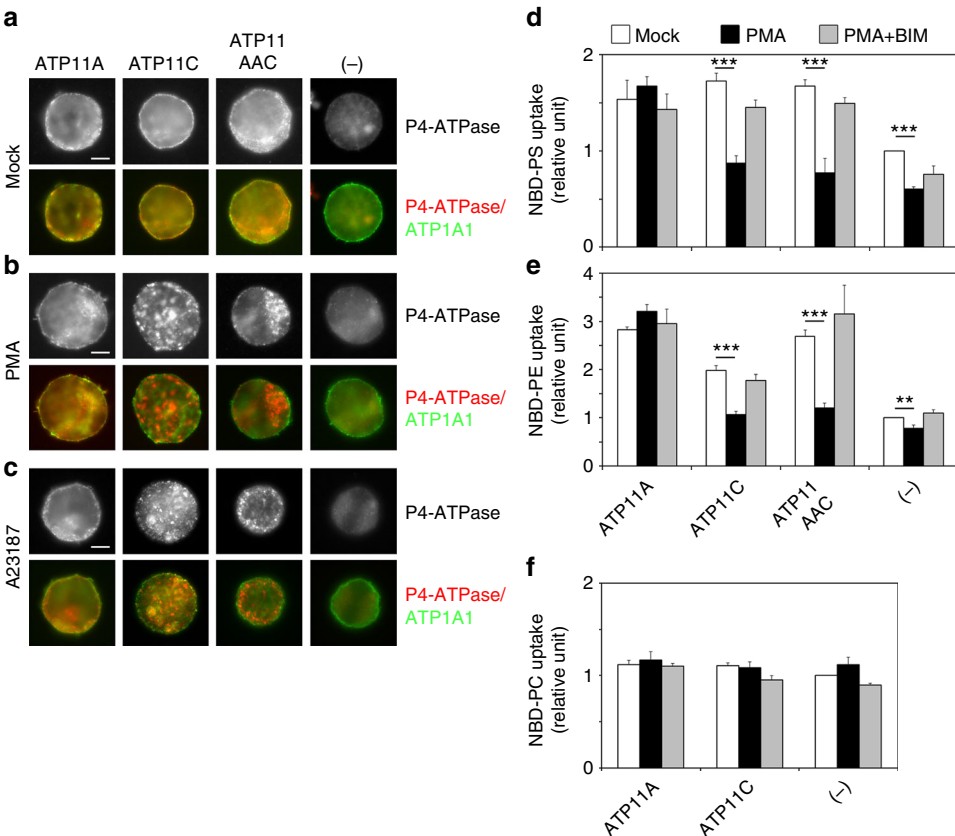

**Fig. 5** C-terminus of ATP11C is required for inhibition of flippase activities by PKC activation. **a**–**c** Ba/F3 cells stably expressing HA-tagged ATP11A, ATP11C, and ATP11AAC chimeric proteins were treated with vehicle alone (**a**), 400 nM PMA (**b**), or 1 μM A23187 in the presence of CaCl$_2$ (**c**) for 15 min. The cells were fixed and immunostained with anti-HA and anti-ATP1A1 antibodies, followed by incubation with Cy3-conjugated anti-rat and Alexa Fluor 488-conjugated anti-rabbit secondary antibodies. Scale bars, 5 μm. **d**–**f** Ba/F3 cells were treated with vehicle alone (white bars), 400 nM PMA (black bars), or 400 nM PMA and 2 μM of BIM-1 (gray bars) for 15 min. The cells were then washed with flippase assay buffer and incubated at 15 °C for 5 min with NBD-PS (**d**), or 15 min with NBD-PE (**e**) or NBD-PC (**f**). After extraction with fatty acid-free BSA, the residual fluorescence intensity associated with the cells was determined by flow cytometry. Fold increase of NBD-lipids uptake is shown relative to that in Mock-treated parental Ba/F3 cells (−). Graphs display averages from three independent experiments ± SD. **p < 0.005; ***p < 0.001

(Fig. 5e), suggesting that the substrate preference of ATP11AAC is similar to that of ATP11A.

**Ser1116 is critical for PMA-induced endocytosis of ATP11C.** The C-terminal cytoplasmic region of ATP11C contains eight serine residues (S1097, S1103, S1108, S1110, S1112, S1116, S1126, and S1129) and one threonine residue (T1124), as shown in Fig. 6a. Among the nine Ser/Thr residues, eight (with the exception of S1097) are registered as phosphorylated sites in the public phosphosite database (PhosphoSitePlus, http://www.phosphosite.org/). Therefore, we investigated the possibility that those Ser/Thr residues in C-terminal region of ATP11C participate in its endocytosis upon PKC activation. To this end, we replaced each Ser/Thr residue with Ala, transiently expressed each Ala mutant in HeLa cells, and observed protein localization following PMA treatment (Supplementary Fig. 3a). When Ser1116 was replaced with Ala (S1116A), PMA-induced internalization of ATP11C was completely blocked (Supplementary Figs 3a and 6b). Mutation of Ser1126 to Ala also inhibited PMA-induced endocytosis, although some fraction of the protein was present in intracellular endosomal structures (Supplementary Figs 3a and 6b). Mutants other than S1116A and S1126A were endocytosed as efficiently as ATP11C(WT) upon PMA treatment (Supplementary Fig. 3a). These results suggest that Ser1116 and Ser1126 are important for PMA-induced endocytosis of ATP11C, although Ser1116 is more critical. Because Ser1116 and Ser1126 can be

phosphorylated[52–54], we generated mutants in which Ser1116 or Ser1126 was replaced with Asp to mimic the phosphorylated state. Strikingly, the S1116D mutant localized to endosomal compartments in the absence of PMA (Fig. 6b, upper panel of S1116D and Supplementary Fig. 3b). These observations demonstrate that phosphorylation at the Ser1116 residue is indispensable for endocytosis of ATP11C, and suggest that this phosphorylation might be mediated by PKC. Although mutation of Ser1126 to Ala partially inhibited endocytosis (Fig. 6b), mutation of Ser1126 to Asp did not affect the localization of ATP11C in the absence of PMA (Fig. 6b, upper panel of S1126D), suggesting that the phosphorylation of Ser1126 residue plays an auxiliary role in PMA-induced ATP11C endocytosis.

Next, we asked whether phosphorylation of Ser1116 is solely responsible for the PMA-induced endocytosis of ATP11C. To this end, we generated four mutants by substituting Ala for all nine Ser/Thr residues (9-Ala); eight residues, excluding Ser1116 or Ser1126 (8-Ala(S1116) or 8-Ala(S1126), respectively); or seven residues, excluding Ser1116 and Ser1126 [7-Ala(S1116/1126)] (Fig. 6c). Upon PMA treatment, the 9-Ala and 8-Ala(S1126) mutants of ATP11C were not internalized (Fig. 6c). By contrast, 8-Ala(S1116) was partially internalized and 7-Ala(S1116/1126) internalized as efficiently as ATP11C(WT) (Fig. 6c). Therefore, phosphorylation of Ser1116 is critical for PMA-induced endocytosis of ATP11C, and phosphorylation of Ser1126 plays an ancillary role.

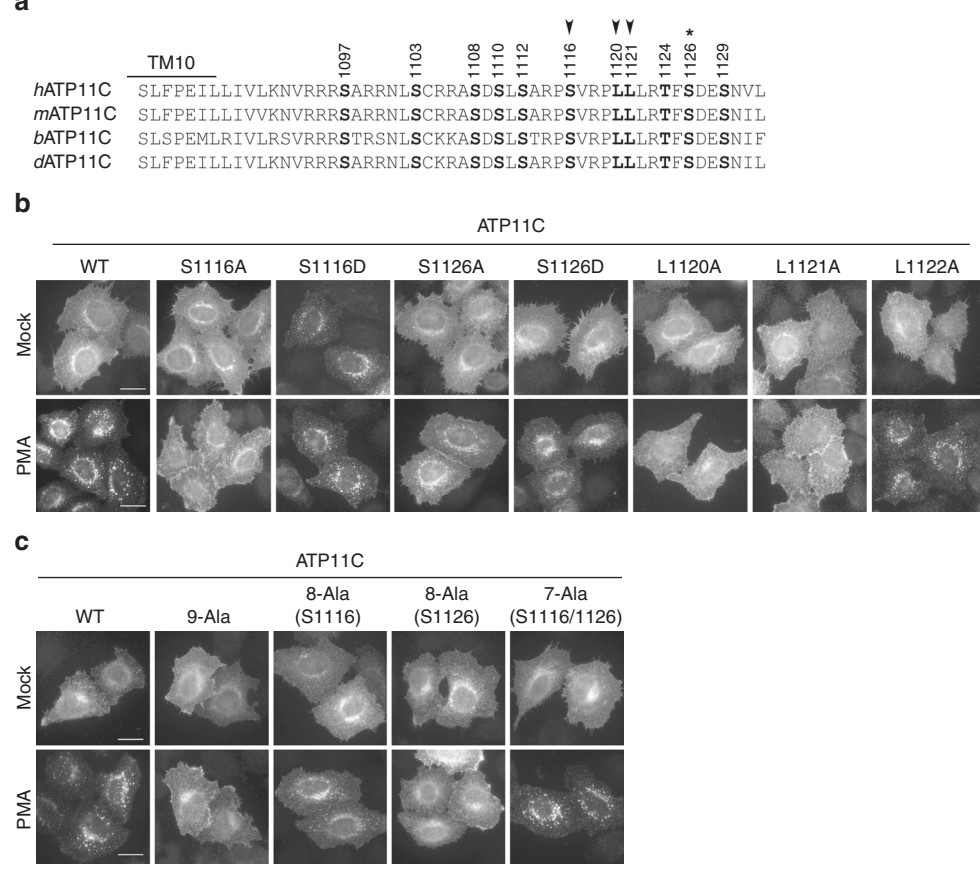

**Fig. 6** Ser1116, Leu1120, and Leu1121 serve as a di-leucine signaling motif upon PKC activation. **a** Alignment of amino acid sequences of the C-terminal cytoplasmic region of mammalian ATP11C. Serine and threonine residues shown in bold were conserved among human (*h*), mouse (*m*), bovine (*b*), and dog (*d*). Arrowheads exhibit critical residues for endocytosis, and the residue marked with an asterisk is involved in endosomal localization of ATP11C after PKC activation. **b** HeLa cells were transiently co-transfected with expression vectors for FLAG-tagged CDC50A and HA-tagged WT of ATP11C, or mutants of HA-tagged ATP11C, in which each bold residue in **a** was replaced with alanine (see also Supplementary Fig. 3a) or aspartate. **c** HeLa cells were transiently co-transfected with expression vectors for FLAG-tagged CDC50A and HA-tagged WT of ATP11C, or mutants of HA-tagged ATP11C. In the mutants, alanine replaced all serine and threonine residues in the C-terminus of ATP11C shown in **a** (9-Ala), all except Ser1116 (8-Ala(S1116)), all except Ser1126 (8-Ala[S1126]), or all except Ser1116 and Ser1126 (7-Ala(S1116/1126)). **b**, **c** The cells were treated with vehicle alone (Mock) or 400 nM of PMA (PMA) for 15 min and fixed. The fixed cells were then incubated with anti-HA antibody, followed by incubation with Cy3-conjugated anti-rat secondary antibody. Scale bars, 20 μm

**Phosphorylation of Ser1116 is required for ATP11C endocytosis.** In light of the potential phosphorylation of Ser1116 by PKC, we noted that the pS1116VRPLL1120/1121 sequence (Fig. 6a) in the C-terminal region of ATP11C is similar to the [DE]XXXL[LI] (di-leucine) motif, which plays critical roles in intracellular trafficking and downregulation (endocytosis) of transmembrane proteins, such as CD3, CD4, GLUT4, and ATP7A/B, a member of the P-type ATPase superfamily[43]. To determine whether the SVRPLL sequence of ATP11C serves as a di-leucine sorting motif, we mutated Leu1120, Leu1121, or Leu1122 to Ala and observed the localization of the mutant proteins in the presence or absence of PMA (Fig. 6b). The L1120A and L1121A mutants remained localized at the plasma membrane upon PMA treatment, whereas the L1122A mutant was endocytosed as ATP11C(WT) (Fig. 6b), indicating that Leu1120 and Leu1121, but not Leu1122, are critical for endocytosis. Therefore, PKC-induced endocytosis of ATP11C requires phosphorylation of Ser1116 (act as an acidic residue) and the di-leucine residues, Leu1120 and Leu1121.

To confirm that the phospho-Ser1116-containing di-leucine sequence (pSVRPLL) is sufficient for endocytosis, we constructed chimeric proteins possessing Lyn-EGFP (plasma membrane-targeting signal of Lyn[55] and EGFP, Fig. 7a), which is fused to the C-terminal cytoplasmic region of ATP11C: CT-WT, CT-9-Ala (9 Ser/Thr to Ala), CT-8-Ala-S1116 (8 Ser/Thr to Ala, except Ser1116), and CT-8-Ala-S1126 (8 Ser/Thr to Ala, except Ser1126) (Fig. 7b). All constructs localized to the plasma membrane in the absence of PMA (Fig. 7Cc, Mock). Lyn-EGFP-CT-WT, but not the original construct Lyn-EGFP-FRB (−), was internalized upon treatment of cells with PMA, indicating that the C-terminal region of ATP11C is sufficient for PMA-induced endocytosis (Fig. 7c, a middle panel). Lyn-EGFP-CT-WT was also internalized following an increase in cytosolic Ca$^{2+}$ by A23187 treatment (Fig. 7c, a bottom panel). Importantly, the Lyn-EGFP-CT-8-Ala-S1116 mutant was internalized as efficiently as Lyn-EGFP-CT-WT upon PMA or A23187 treatment (Fig. 7c). By contrast, the Lyn-EGFP-CT-9-Ala or Lyn-EGFP-CT-8-Ala-S1126 mutant was not internalized upon treatment of cells with either PMA or A23187 (Fig. 7c), although the di-leucine residues (Leu1120/1121) were present in these constructs (Fig. 7b). These results demonstrate that the Ser1116 phosphorylation in the C-terminal region of ATP11C by PKC creates a functional di-leucine motif that is sufficient for endocytosis.

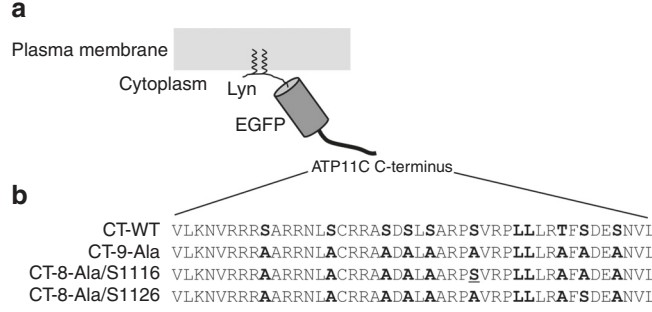

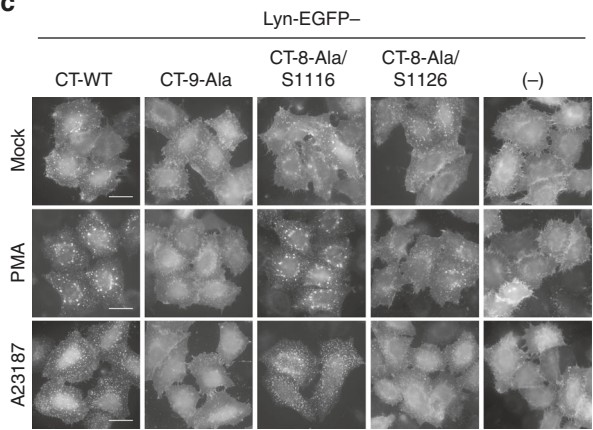

**Fig. 7** Phosphorylation of Ser1116 containing the Leu1120/Leu1121 (pSVRPLL) motif is sufficient for endocytosis. **a** Schematic representation of chimeric protein. The plasma membrane targeting signal of Lyn and EGFP was fused to the C-terminal cytoplasmic region (CT) of ATP11C. **b** Amino acid sequences in the C-terminal cytoplasmic region (CT) of human ATP11C, which were fused to Lyn-EGFP as shown in **a**. **c** HeLa cells were transiently transfected with expression vector for Lyn-EGFP-tagged CT-WT, CT-9-Ala, CT-8-Ala/S1116, or CT-8-Ala/S1126, or the original vector encoding Lyn-EGFP-tagged FRB (FKBP–rapamycin-binding domain). The cells were treated for 15 min with vehicle alone (Mock), 400 nM PMA (PMA), or 1 μM A23187 in the presence of 1.8 mM CaCl$_2$ (A23187), followed by fixation. Scale bars, 20 μm

**Regulation of flippase activity of ATP11C in the plasma membrane by endocytosis**. Next, we examined the PS-flippase activity of endocytosis-defective ATP11C mutants (S1116A, L1120A, and L1121A). To this end, we established Ba/F3 cells stably expressing the mutants, as well as the L1122A mutant as a negative control. Consistent with the results of transient expression in HeLa cells, the S1116A, L1120A, and L1121A mutants were not endocytosed, whereas ATP11C(WT) and the L1122A mutant were efficiently endocytosed upon PMA or A23187 treatment of Ba/F3 cells (Fig. 8a, b). We then assessed the cell-surface level of ATP11C and its mutants, as well as ATP11A, by surface biotinylation analysis. The cell surface levels of ATP11C (WT) and the L1122A mutant on the cell surface decreased upon PMA treatment, whereas those of endocytosis-defective mutants did not substantially decrease (Fig. 8c, d), and the surface level of ATP11A actually increased.

The PS-flippase activities of cells stably expressing ATP11C mutants was comparable to that of ATP11C(WT) under basal conditions (Fig. 8e, white bars), indicating that these mutations did not affect the enzymatic activity of ATP11C. Upon treatment with PMA, uptake of NBD-PS was dramatically decreased in cells expressing ATP11C(WT) and ATP11C(L1122A), but not in those expressing ATP11A (Fig. 8e, black bars). Interestingly, cells expressing endocytosis-defective mutants (S1116A, L1120A, and

L1121A) also exhibited reduced PS-flippase activity to some extent (Fig. 8e). When the ratio of PS-flippase activities in the presence vs. absence of PMA was estimated as shown in Fig. 8f, the ratio was decreased by >50% in cells expressing ATP11C (WT) and ATP11C(L1122A), and to a lesser extent in cells expressing the endocytosis-defective mutants, S1116A, L1120A, and L1121A. By contrast, the ratio was not significantly altered in ATP11A-expressing cells. To investigate the involvement of other C-terminal phosphorylation sites of ATP11C in PS uptake, we tested PS-flippase activity in Ba/F3 cells stably expressing the ATP11C(9-Ala) mutant (Fig. 8g); the ratio of the PS-flippase activities in the presence vs. absence of PMA is shown in Fig. 8h. PS uptake was slightly inhibited by PMA treatment in cells expressing the ATP11C(9-Ala) mutant, although to a greater extent to that in cells expressing ATP11C(WT) (Fig. 8h). Taken together, these findings indicate that sequestration of ATP11C from the plasma membrane by endocytosis plays a role in the efficient down-regulation of the enzymatic activity of ATP11C, although the activity might also be partially inhibited by PMA treatment via an unknown mechanism.

**Internalization of ATP11C is dependent on clathrin and PKCα**. Because the di-leucine motif is recognized by the clathrin adaptor protein complex AP-2[44,56], we next asked whether ATP11C endocytosis is mediated by a clathrin-dependent pathway. To this end, we knocked down clathrin heavy chain by RNAi in HeLa cells stably expressing ATP11C, and then treated the cells with PMA or A23187 (Fig. 9). The efficient depletion of clathrin heavy chain was confirmed by immunoblotting and immuno-fluorescence analyses (Supplementary Figs 4a and 9a, b, bottom panels). Internalization of ATP11C induced by PMA or an increase in cytosolic Ca$^{2+}$ by A23187 treatment was dramatically inhibited by depletion of clathrin heavy chain in comparison with the control (Fig. 9b), indicating that ATP11C undergoes clathrin-mediated endocytosis upon PKC activation. We classified and counted cells with ATP11C only at the plasma membrane, both at the plasma membrane and in intracellular compartments, and only in intracellular compartments (Fig. 9d–f). In control cells, ATP11C was endocytosed and did not appear at the plasma membrane upon treatment of cells with PMA or A23187, although endocytosis was induced less efficiently by A23187 than PMA (Fig. 9d). By contrast, in cells depleted of clathrin heavy chain, ATP11C mostly remained at the plasma membrane upon PMA or A23187 treatment (Fig. 9b, e). These results indicate that ATP11C is internalized by clathrin-mediated endocytosis, probably via its interaction with AP-2 through the di-leucine motif, which is generated by signal-dependent phosphorylation of Ser1116.

We then sought to determine which PKC isoform is involved in the ATP11C internalization. To address this question, we knocked down various PKC isoforms (Supplementary Fig. 4b, d) by RNAi. Because the signal-dependent endocytosis of ATP11C was Ca$^{2+}$-dependent, we speculated that cPKC (α, β, and γ) might be involved. However, we knocked down not only cPKC (PKCα, and β) but also PKCδ, ε, and ζ (Supplementary Fig. 4b, d); we did not include PKCγ in this experiment because it is specifically expressed in central nervous system. Furthermore, PKCβ was not expressed in HeLa cells, as determined by RT-PCR analysis (Supplementary Fig. 4c). Therefore, only the PKCα isoform among cPKC is expressed in HeLa cells. As shown in Supplementary Fig. 4d, Fig. 9c, f, depletion of PKCα, but not other isoforms, inhibited the internalization of ATP11C upon treatment of cells with A23187. However, knockdown of PKCα did not inhibit the ATP11C internalization in response to PMA (Fig. 9c, f); because PMA is a potent and irreversible PKC

activator, residual PKCα might facilitate the ATP11C internalization. Taken together, these observations indicate that the $Ca^{2+}$-dependent endocytosis of ATP11C is mediated by PKCα activation.

**ATP11C is internalized by stimulation of Gq-coupled receptor.** Next, we asked whether the internalization of ATP11C takes place under physiologically relevant conditions. Firstly, we established HeLa cells stably co-expressing Gq-coupled serotonin receptor (5-hydroxytryptamine (HT) receptor 2A; 5-HT2A-R) and

ATP11C-HA (Fig. 10c, d). Stimulation of Gq-coupled receptors activates phospholipase C, which catalyzes hydrolysis of phosphatidylinositol 4,5-bisphosphate to inositol 1,4,5-trisphosphate ($IP_3$) and diacylglycerol (DAG). $IP_3$ triggers the release of $Ca^{2+}$ from intracellular stores and PKC is activated by cytosolic $Ca^{2+}$ and DAG (Supplementary Fig. 6). As shown in Fig. 10c, ATP11C was internalized following treatment of the cells co-expressing 5-HT2A-R and ATP11C with serotonin (5-HT) for 10 min, but not in cells expressing ATP11C alone (Fig. 10a), indicating that $Ca^{2+}$ signaling via Gq-coupled 5-HT2A-R induced the ATP11C

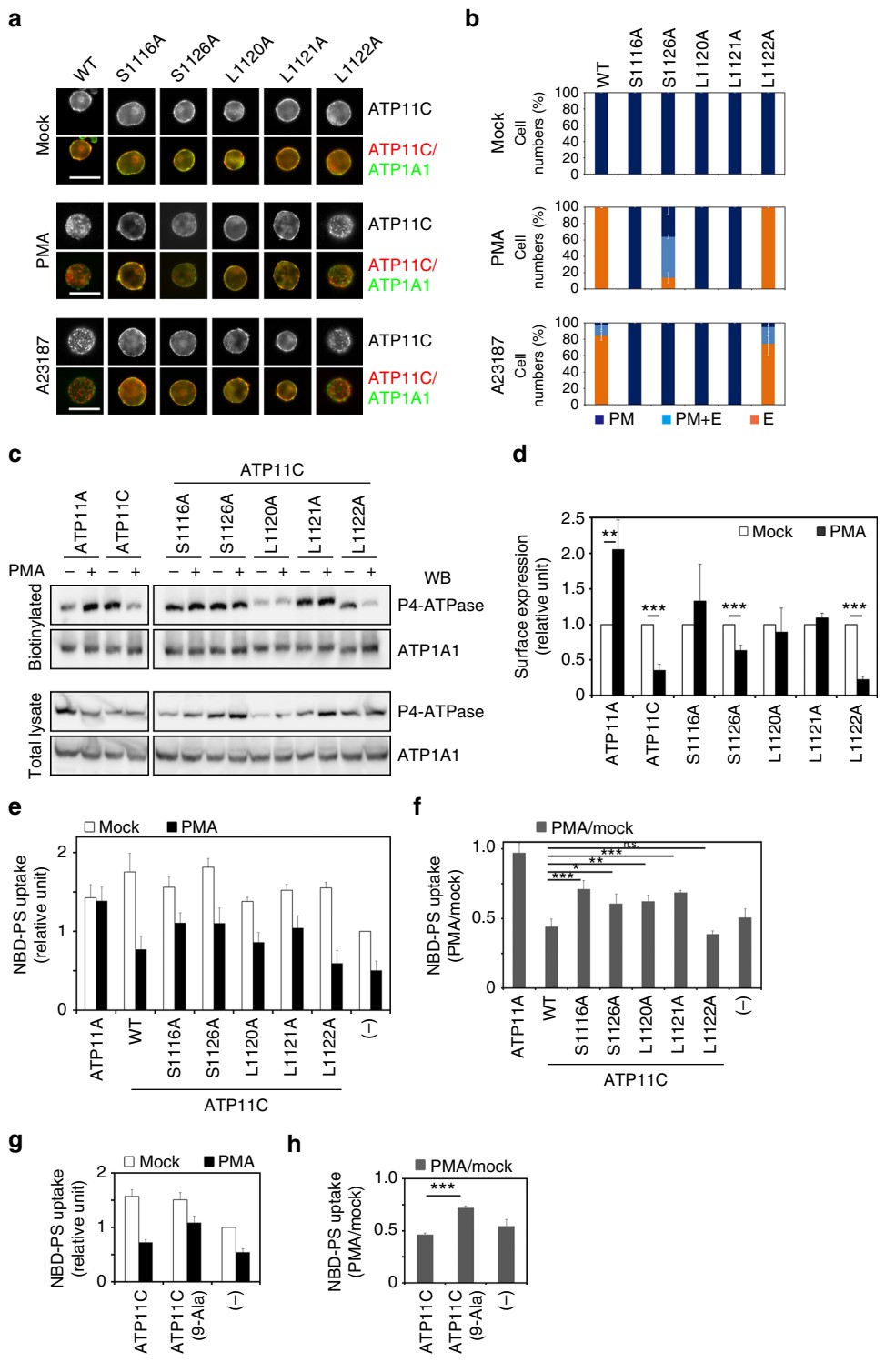

endocytosis. To confirm that 5-HT2A-R-mediated $Ca^{2+}$ signaling gives rise to the ATP11C endocytosis, we observed the ATP11C endocytosis in cells treated with BAPTA-AM to chelate cytosolic $Ca^{2+}$. As shown in Supplementary Fig. 5a, the endocytosis of ATP11C induced by 5-HT treatment was dramatically inhibited by pretreatment of cells with BAPTA-AM, demonstrating that the ATP11C endocytosis is elicited by an increase in the cytosolic $Ca^{2+}$ level in response to 5-HT. The ATP11C endocytosis induced by 5-HT treatment also occurred in cells stably co-expressing C-terminally FLAG-tagged 5-HT2A-R and ATP11C-HA (Supplementary Fig. 5b, c). It is noteworthy that the FLAG-tagged 5-HT2A-R was also endocytosed[57] and colocalized with ATP11C upon 5-HT treatment (Supplementary Fig. 5). Internalized ATP11C was colocalized with EEA1 but not with Lamp-1 (Fig. 10c, enlarged panels), consistent with the results shown in Fig. 3. Furthermore, when the cells washed and replaced with 5-HT-free medium, endosomal ATP11C disappeared and the plasma membrane localization of ATP11C was restored (Fig. 10d). When the cells were classified into those with ATP11C only at the plasma membrane and those in intracellular compartments as well as at the plasma membrane (endocytosed) (Fig. 10b, e), the population of cells with endosomal ATP11C was dramatically increased upon 5-HT treatment for 10 min, while removal of 5-HT caused return of ATP11C to the plasma membrane (Fig. 10e) probably due to a reduction in the cytosolic $Ca^{2+}$ level. The 5-HT-induced ATP11C internalization was not observed in control cells, which did not express exogenous 5-HT2A-R (Fig. 10b). Recycling of ATP11C probably requires SNX27 and Vps35, a component of the retromer complex[58]; consistent with the observation, when cells were treated with PMA or A23187, internalized ATP11C was extensively colocalized with Vps35 (Fig. 2b, c, enlarged insets). We then visualized the endocytosis of EGFP-tagged ATP11C in cells expressing 5-HT2A-R by time-lapse imaging analysis (Supplementary Movie 2). EGFP-ATP11C was localized to the plasma membrane just after serum starvation, and rapidly internalized and accumulated to endosomal compartments upon 5-HT treatment (Supplementary Movie 2). We also investigated the ATP11C endocytosis via $Ca^{2+}$ signaling mediated by an endogenous Gq-coupled receptor. HeLa cells are known to undergo $Ca^{2+}$ signaling upon stimulation with histamine via Gq-coupled histamine H1 receptor (H1-R)[59–61]. ATP11C was internalized following treatment of cells with histamine for 10 min (Fig. 10f), although to a lesser extent compared with that induced by PMA or A23187 treatment, indicating that $Ca^{2+}$ signaling via endogenous H1-R induced the ATP11C endocytosis. Internalized ATP11C was colocalized with EEA1 but not with Lamp-1 (Fig. 10f, enlarged panels), consistent with the results shown in Figs. 3 and 10c.

When the cells were classified into those with ATP11C only at the plasma membrane and those in intracellular compartments as well as at the plasma membrane (endocytosed) (Fig. 10g), the endosomal ATP11C level was increased upon treatment of cells with histamine for 10 min, while a longer histamine treatment (60 min) causes return of ATP11C to the plasma membrane (Fig. 10f, g), consistent with the results shown in Fig. 10d.

## Discussion

In this study, we demonstrated that internalization of ATP11C is induced by $Ca^{2+}$-dependent PKCα activation, and revealed that phosphorylation of Ser1116 in the C-terminal cytoplasmic region of ATP11C results in generation of a functional di-leucine motif (pSVRPLL) required for endocytosis. Furthermore, the ATP11C internalization is mediated by clathrin-dependent endocytosis, and is responsible for the decrease in the PS-flipping activity at the plasma membrane. 5-HT2A-R and H1-R (Gq-coupled receptors)-mediated $Ca^{2+}$ signaling enabled the endocytosis of ATP11C, and thus the signal-dependent endocytosis of ATP11C constitutes an important physiological mechanism for regulation of local asymmetrical PS distribution in the plasma membrane (Supplementary Fig. 6). Downregulation of ATP11C and activation of $Ca^{2+}$-dependent scramblase may enable cell-surface exposure of PS at specific times and places (Supplementary Fig. 6). The mechanism of ATP11C downregulation seems to be conserved in adherent (HeLa) and non-adherent (pro-B cell line Ba/F3) cells. Since ATP11C deficiency causes a defect in B-cell maturation, altered erythrocyte shape and anemia[25,26,62], it is tempting to speculate that PKC-mediated regulatory mechanism of ATP11C contributes to B-cell differentiation or erythrocyte homeostasis.

It is noteworthy that knockout of ATP11C does not result in the cell surface exposure of PS (our unpublished data)[23], although flippase activity toward NBD-PS is dramatically decreased. Contrary to knockout of ATP11C, that of CDC50A leads to PS exposure (our unpublished data)[23], indicating that global asymmetric transbilayer distribution of PS could be maintained by multiple PS-specific P4-ATPases. Moreover, there might be another compensatory mechanism to prevent PS-exposure. In fact, upon treatment of cells with PMA, the surface level of ATP11A was increased (Figs. 1b and 8c), and may compensate for PMA-induced inhibition of PS-flippase activity of ATP11C, although we do not know the mechanism and physiological relevance of the alteration of ATP11A level.

The AP-2 adaptor complex (composed of α, β2, μ2, and σ2 subunits) recognizes two distinct endocytic motifs, YXXΦ (Φ stands for a bulky hydrophobic residue) and [DE]XXXL[LI].

**Fig. 8** Flippase activity of ATP11C inhibited by PKC is partially rescued by endocytosis-defective mutants. **a–d** Ba/F3 cells stably expressing the indicated point mutants of ATP11C-HA were treated with vehicle alone (Mock), 400 nM PMA (PMA), or 1 μM A23187 in the presence of CaCl$_2$ (A23187) for 15 min. **a** Cells were immunostained with anti-HA and anti-ATP1A1 antibodies, followed by incubation with Cy3-conjugated anti-rat and Alexa Fluor 488-conjugated anti-rabbit secondary antibodies. Scale bars, 20 μm. **b** The cells with ATP11C mutants localized to the plasma membrane (PM), to the plasma membrane and endosomes (PM + E), or to the endosomes (E) were counted; counts were normalized against the total number of counted cells. In each sample, 315–459 cells were counted. Graphs display averages ± SD from three independent experiments. **c** Cell-surface expression level of ATP11A, ATP11C, and the indicated mutants upon treatment with PMA were analyzed after surface biotinylation. Proteins precipitated with streptavidin-agarose beads were subjected to immunoblot analysis (upper panels, biotinylated). Ten percent of the input of the biotinylation reaction was loaded in each lane (lower panels, total lysate). Expression levels of P4-ATPases were analyzed by immunoblotting with anti-HA and anti-ATP1A1 (as an internal control) antibodies. **d** Relative surface expression levels of proteins, normalized against the level of ATP1A1 (used as an internal control). Graphs display averages ± SD from four independent experiments. **p < 0.005; ***p < 0.001. **e, g** Ba/F3 cells stably expressing ATP11A, ATP11C, the indicated point mutant of ATP11C, or ATP11C(9-Ala) were treated with vehicle alone (white bars) or 400 nM PMA (black bars) for 15 min. The cells were then washed with flippase assay buffer and incubated with NBD-PS at 15 °C for 5 min. After extraction with fatty acid-free BSA, the residual fluorescence intensity associated with the cells was determined by flow cytometry. Fold increase in NBD-PS uptake is shown relative to that in parental Ba/F3 cells (−). Graphs display averages ± SD from four (**e**) or three (**g**) independent experiments. **f, h** Ratio of PS-flip activity in PMA vs. mock-treated cells in **e** and **g**, respectively. Graphs display averages ± SD from four **f** or three **h** independent experiments. *p < 0.01; **p < 0.005; ***p < 0.001

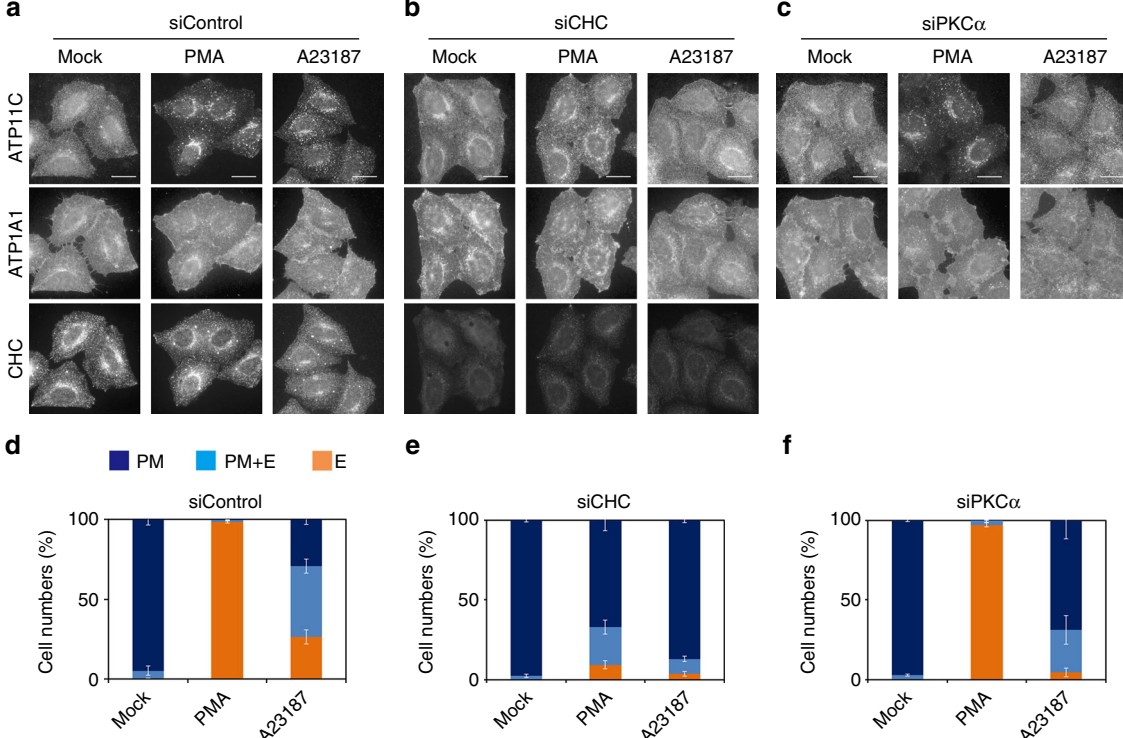

**Fig. 9** ATP11C is downregulated by clathrin-mediated endocytosis upon PKC activation. HeLa cells stably expressing ATP11C-HA were transfected with non-targeting siRNA (siControl) (**a**, **d**) or siRNAs against clathrin heavy chain (CHC) (**b**, **e**) or PKCα (**c**, **f**). (see Supplementary Fig. 4). Each siRNA-treated sample was treated with vehicle alone (Mock), 400 nM PMA (PMA), or 1 μM A23187 in the presence of CaCl₂ (A23187) for 15 min, followed by fixation. The fixed cells were permeabilized and incubated with anti-HA, anti-ATP1A1, and anti-CHC antibodies, followed by Cy3-conjugated anti-rat, Alexa Fluor 647-conjugated anti-rabbit, and Alexa Fluor 488-conjugated anti-mouse secondary antibodies. Scale bars, 20 μm. **d–f** The cells in which ATP11C localized to the plasma membrane (PM), to the plasma membrane and endosomes (PM + E), or to endosomes (E) were counted; counts were normalized against the total number of counted cells. In each sample, 643–783 cells were counted. Graphs display averages ± SD from three independent experiments

The former tyrosine-containing motif binds to the binding pocket of μ2, whereas the latter di-leucine motif binds to that formed by α and σ2[44]. The YXXΦ and [DE]XXXL[LI] signals play critical roles in the sorting of many transmembrane proteins: CI-MPR, furin, TGN46, CD63, and Lamp-1 contain the YXXΦ motif; and GLUT4, CD3, CD4, and ATP7A/B (a Cu⁺ transporter) contain the [DE]XXXL[LI] motif[43]. T-cell receptor and the T-cell surface antigen CD4 are internalized from the plasma membrane via clathrin-mediated endocytosis after PKC-mediated receptor phosphorylation[63,64]. CD4 undergoes a similar downregulation upon phosphorylation of a serine residue within its SQIKRLL (SQXXXLL) sequence, although it contains Q instead of D/E at the −4 position from the di-leucine residues[65]. The DKQTLL (DXXXLL) sequence of CD3γ, a T-cell receptor (TCR) subunit, is part of an adjustable SDKQTLL signal that participates in serine phosphorylation-dependent downregulation of the TCR from the cell surface[64]. Interestingly, ATP11C contains a SVRPLL sequence in its C-terminal cytoplasmic region that cannot serve as an endocytic signal unless the Ser residue is phosphorylated in response to cellular signals. Thus, endocytosis of ATP11C is tightly regulated, which is reasonably given that ATP11C plays an important role in maintaining asymmetric PS distribution at the plasma membrane[19,23,22]. On the other hand, PMA-treatment slightly decreased the flippase activity of ATP11C(9-Ala) (Fig. 8g), suggesting that the ATP11C activity, regardless of its endocytosis, could be marginally inhibited. Phosphorylation of ATP11C besides its C-terminal region, or interaction of ATP11C with other phosphorylated proteins might contribute to the inhibition. Although ATP11A also contains a STIFML sequence, which is similar to dileucine-based signals, in its C-terminal

cytoplasmic region (Supplementary Fig. 2), it could not serve as the endocytic signal at least by Ca²⁺-mediated PKC activation.

ATP11C may cycle between the plasma membrane and endosomes, because its cell-surface expression is reduced in SNX27- or VPS35-depleted cells[58]. Thus, SNX27 and the retromer complex regulate the recycling pathway of ATP11C from endosomes to the plasma membrane. When cells are exposed to a receptor-mediated signal leading to PKC activation, phosphorylation of ATP11C allows its removal from the plasma membrane, which may contribute to the disruption of asymmetric PS distribution in the plasma membrane. Internalized ATP11C is not transported to lysosomes for degradation (Fig. 3), suggesting that it may be recycled back to the plasma membrane soon after termination of signaling, thus contributing to homeostasis of PS distribution at the plasma membrane. Indeed, treatment of cells with 5-HT or histamine once causes internalization of ATP11C, and further incubation restores the plasma membrane localization of ATP11C (Fig. 10). When and where is signal-dependent downregulation of ATP11C required? Spatiotemporally regulated PS exposure occurs not only in cells fated for elimination, such as platelets and apoptotic cells, but also in living cells, such as activated immune cells, capacitated sperm cells, and mast and neuroendocrine cells during signal-dependent exocytosis[31–33,45,66–68]. Therefore, signal-dependent downregulation of ATP11C might participate in temporal redistribution of PS at the plasma membrane (Supplementary Fig. 6). Because apoptotic cells do not need to restore asymmetric distribution of PS, ATP11A and ATP11C are cleaved by caspases, concomitant with activation of the phospholipid scrambling[23,69]. Is downregulation of ATP11C required for PS exposure in

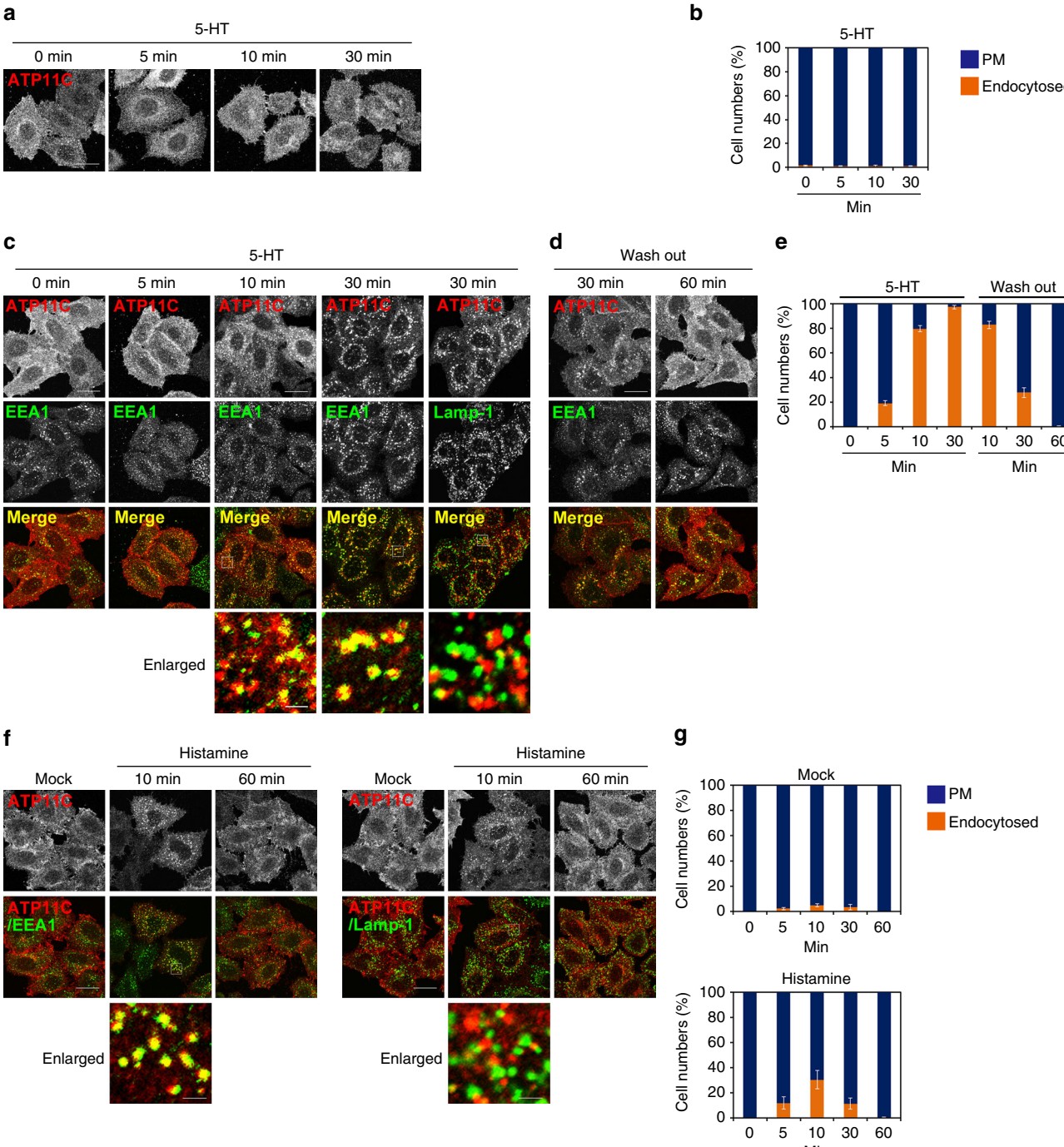

**Fig. 10** ATP11C is endocytosed by treatment with serotonin or histamine. **a**–**e** HeLa cells stably expressing C-terminally HA-tagged ATP11C alone (**a**, **b**) or ATP11C-HA and untagged 5-HT2A-R (**c**–**e**) were serum starved for 3 h and treated with 500 nM 5-HT for indicated times **c**. See Supplementary Movie 2. After 30 min treatment, cells were washed with 5-HT-free medium and incubated in the medium for indicated times (**d**). **f**, **g** HeLa cells stably expressing C-terminally HA-tagged ATP11C were treated with vehicle alone (Mock) or 5 μM of histamine for indicated times. The cells were fixed and doubly stained for HA, and EEA1 or Lamp-1 followed by incubation with Cy3-conjugated anti-rat and Alexa Fluor 488-conjugated anti-mouse secondary antibodies (**f**). Images were obtained by confocal microscopy. Scale bars, 20 μm. Insets are enlarged (bars, 2 μm). **b**, **e**, **g** Cells in which ATP11C localized to the plasma membrane (PM), or to the plasma membrane and endosomes (endocytosed) were counted; counts were normalized against the total number of counted cells. In each sample, 370–1273 cells were counted. Graphs display averages ± SD from four (**b**, **e**) or three (**g**) independent experiments

addition to activities of scramblases? Although the activation of scramblases might be enough for initial PS exposure, stable PS exposure for further cellular functions and/or PS-selectivity would be attributed to the inhibition of ATP11C since scramblases do not seem to have a substrate specificity[39]. Notably, during apoptosis induction, caspase-resistant ATP11C did not allow PS exposure, suggesting that inhibition of flippase as well as activation of scramblase is most likely required for PS exposure[4,23,70].

Recent studies have suggested that choleretic and cholestatic effects may be mediated by PKC isoforms[71]. PKCα mediates retrieval of the bile salt exporting pump (BSEP/ABCB11) induced

by PMA[72], and PKCδ mediates the cAMP-induced translocation of sodium–taurocholate co-transporting polypeptide (NTCP) and multidrug resistance-associated protein 2 (MRP2) to the plasma membrane in rat hepatocytes[73]. ATP11C deficiency causes hyperbilirubinemia and hypercholanemia in mice[27], and ATP11C appears to be involved in transport of bile acids across the sinusoidal membrane[74,75]. Therefore, ATP11C might be important for the regulation of bile acid transporters in the sinusoidal membrane of hepatocytes. PKC isoform-specific signaling may regulate not only BSEP and MRP2 in the canalicular membrane, but also ATP11C and NTCP in the sinusoidal membrane, to exert a choleretic effect.

## Methods

**Plasmids.** Expression vectors for C-terminally HA-tagged human P4-ATPases were constructed as described previously[9,10]. An expression vector for N-terminally EGFP-tagged ATP11C was constructed by subcloning a cDNA fragment containing the entire coding sequence of human ATP11C into pEGFP-C1 (Invitrogen) or pCAG-EGFP. To construct pCAG-EGFP, DNA fragment encoding EGFP was amplified from pEGFP-C1 and inserted into pCAG vector. Chimeric ATP11A with the N- or C-terminal cytoplasmic region from ATP11C and chimeric ATP11C with the N- or C-terminal cytoplasmic region from ATP11A were constructed using PCR and the In-Fusion HD cloning kit (Clontech). Point mutations of ATP11C were introduced into the ATP11C cDNA using the QuikChange II XL site-directed mutagenesis kit (Agilent Technologies). For construction of ATP11C(9-Ala), ATP11C(8-Ala(S1116)), and ATP11C(8-Ala(S1126)), DNA fragments encoding the C-terminal region of ATP11C with the indicated mutations (135 base pairs) were synthesized (Thermo Fisher Scientific) and inserted to replace the same region of WT ATP11C cDNA using the SLiCE cloning method[76]. For construction of ATP11C(7-Ala(S1116/S1126), a point mutation (Ala1126 to Ser) was introduced into the ATP11C(8Ala(S1116)) cDNA using the QuikChange II XL site-directed mutagenesis kit. Lyn-EGFP-FRB (FKBP–rapamycin-binding domain) construct[55] was a kind gift from Takanari Inoue (Johns Hopkins University). For construction of Lyn-EGFP-fused chimeras, DNA fragments encoding the C-terminal region of ATP11C (WT, 9-Ala, 8-Ala(S1116), and 8-Ala(S1126)])were amplified by PCR and inserted to replace the FRB cDNA using BsrGI and XhoI sites. Oligos are listed in Supplementary Table 1. Mouse 5-HT2A receptor cDNA was obtained from mouse (C57BL/6) brain total RNA by RT-PCR using Superscript III First-Strand Synthesis System (Invitrogen) and KOD FX Neo DNA polymerase (Toyobo). The following primer pairs were used: forward, 5′-GCCGGATCCACCATG-GAAATTCTCTGTGAAGACAATATCTC-3′, and reverse, 5′-CGCGTCGACT-CACACACACGTAACCTTTTCATTCACGG-3′ or 5′-CGCGTCGACACAC ACGTAACCTTTTCATTCACGG (for C-3′-terminal tagging). 5-HT2A receptor cDNA was inserted into the pENTR3C vector (Invitrogen). Transfer of the cDNA to pMXs-puro-DEST-FLAG was performed using Gateway system (Invitrogen).

**Antibodies and reagents.** Sources of antibodies used in this study were as follows: monoclonal rabbit anti-ATP1A1 (EP1845Y), Abcam; monoclonal mouse anti-EEA1 (clone14), anti-Lamp1 (H4A3), and anti-clathrin heavy chain (clone23), BD Biosciences; monoclonal mouse anti-TfnR (H68. 4), Zymed; monoclonal mouse anti-clathrin heavy chain (X22), Thermo Fisher Scientific; monoclonal mouse anti-β-tubulin (KMX-1), Millipore; polyclonal rabbit anti-TGN46, kind gift from Minoru Fukuda (Burnham Institute); polyclonal goat anti-VPS35, IMGENEX; monoclonal rat anti-HA (3F10), Roche Applied Science; Alexa Fluor-conjugated secondary antibodies, Invitrogen; Cy3- and horseradish peroxidase-conjugated secondary antibodies, Jackson ImmunoResearch Laboratories (Supplementary Table 2). The NBD-labeled phospholipids (Avanti Polar Lipids) used in this study were NBD-PS (1-oleoyl-2-(6-((7-nitro-2–1,3-benzoxadiazol-4-yl)amino)hexanoyl)-sn-glycero-3-phosphoserine), NBD-PE (1-oleoyl-2-(6-((7-nitro-2-1,3-benzox-adiazol-4-yl)amino)hexanoyl)-sn-glycero-3-phosphoethanolamine), and NBD-PC (1-oleoyl-2-(6-((7-nitro-2-1,3-benzoxadiazol-4-yl)amino)hexanoyl)-sn-glycero-3-phosphocholine). The calcium ionophore A23187, phorbol 12-myristate 13-acetate (PMA), and bisindolylmaleimide I (BIM-1) were purchased from Cayman Chemical, LC Laboratories, and Santa Cruz Biotechnology, respectively. 5-hydroxytryptamine, histamine, and BAPTA-AM were purchased from Nacalai Tesque.

**Cell culture and establishment of stable cell lines.** The interleukin (IL-3)-dependent murine pro-B cell line Ba/F3 and cells expressing recombinant mouse IL-3 were kind gifts from Shigekazu Nagata (Osaka University). The Ba/F3 cells were maintained in RPMI-1640 medium containing 10% fetal calf serum, as described previously[77]. HeLa cells were cultured as described previously[10]. Ba/F3 and HeLa cells expressing each C-terminally HA-tagged P4-ATPase were established as described previously[9]. To transiently express P4-ATPases, HeLa cells were transfected with a pCAG-HA-based vector carrying P4-ATPase cDNA (C-terminally HA-tagged) or pCAG-EGFP containing the ATP11C cDNA (N-terminally EGFP-tagged) and a pcDNA3-FLAG-based vector containing CDC50A cDNA[10]. Transfections were performed by lipofection using X-tremeGENE9 (Roche Applied

Science). One or two days later, the transfected cells were fixed for immunofluorescence analysis. The establishment of HeLa cells co-expressing either untagged 5-HT2A-R or C-terminally FLAG-tagged 5HT2A-R and ATP11C-HA, recombinant retrovirus for expression of 5-HT2A-R or 5-HT2A-R-FLAG was produced and used to infect ATP11C-HA-expressing HeLa cells[9]. The infected cells were selected in medium containing puromycin (1 μg ml⁻¹). A mixed population of drug resistant cells was used for immunofluorescence analysis.

**Immunofluorescence analysis and time-lapse imaging analysis.** Immunostaining was performed as described previously[78] and visualized using an Axiovert 200MAT microscope (Carl Zeiss) and an A1R-MP confocal laser-scanning microscope (Nikon). For time-lapse recording, EGFP-ATP11C expressing HeLa cells were placed on a microscope stage prewarmed to 37 °C, and then observed on an A1R-MP confocal microscope. After 5 min recording, PMA was administered, and images were recorded every 7.8 s for a total of 34 min (Supplementary Movie 1). For Supplementary Movie 2, the cells were incubated in serum-free medium containing 2% BSA for 3 h. After 5 min recording, 5-HT was administered and images were recorded every 8.0 s for a total 79 min.

**RNA interference.** Knockdown of clathrin heavy chain (CHC) or individual PKC isoforms was performed as described previously[14,79]. siRNA oligonucleotides were synthesized by Sigma Genosys as described previously for CHC[80], PKCα[81], and PKCβ, PKCδ, PKCε, and PKCζ[82]. Control siRNA was purchased from Dharmacon. The cells were transfected with siRNAs using Lipofectamine 2000 (Invitrogen) and incubated for 24 h. The transfected cells were then transferred to a culture dish containing coverslips, incubated for an additional 48 h, and processed for immunofluorescence, immunoblot analyses, and RT-PCR analyses as described previously[14,20].

**RT-PCR.** Total RNA was extracted from cells using the RNeasy kit (Qiagen) and subjected to RT-PCR analysis using the SuperScript III One-Step RT-PCR system (Invitrogen). Fragments of PKC isoforms were amplified using the following primer pairs: PKCα: forward, 5′-GGCCAGTGGATGGTACAAGTTG-3′, reverse, 5′-CTGTCGGCAAGCATCACCTTTC-3′;
PKCβ I and II: forward, 5′-GAAAGCCAGTGTTGATGGCTG-3′, reverse, 5′-GCCTTTTCGTTCTGAAAGCATG-3′; PKCδ: forward, 5′-GTGAAGACTGC GGCATGAATG-3′, reverse, 5′-CAAGCAGCACCTTCCCGAAG; PKCε–3′; forward, 5′-CGAGGACTGGATTGATCTGGAG-3′, reverse, 5′-CAGGTGCAGA CTTGACACTGG-3′; PKCζ: forward, 5′-CAGGAGAGCGTACTGCGGTC-3′, reverse, 5′-CTGGCTTAAGGTCCTCCGAG-3′.

**Surface biotinylation.** Cell-surface biotinylation assay was performed as described previously[83]. Briefly, cells were washed three times with chilled PBS containing 0.1 mM CaCl₂ and 0.1 mM MgCl₂ (PBS++), and then incubated at 4 °C for 30 min with 2 mM sulfo-NHS-LC-biotin (Thermo Scientific) in PBS++. To stop biotinylation, the cells were washed three times with chilled PBS++ containing 100 mM glycine and 0.3% BSA, and then washed twice more with chilled PBS. The cells were then lysed in lysis buffer (20 mM HEPES-KOH (pH 7.4) containing 1% NP-40, 150 mM NaCl, and protease inhibitor mixture (Complete, Roche Applied Science) for 30 min at 4 °C. The lysates were centrifuged at maximum speed at 4 °C for 20 min in a microcentrifuge to remove cellular debris and insoluble materials. To precipitate the biotinylated proteins, the supernatant was incubated at 4 °C for 4 h with streptavidin-agarose beads (Thermo Scientific) pre-equilibrated with lysis buffer. The protein-bound streptavidin beads were washed three times with lysis buffer, twice with high-salt buffer (20 mM HEPES-KOH (pH 7.4), 500 mM NaCl, 1 mM EDTA, 0.5% NP-40), and once with 20 mM HEPES-KOH, pH 7.4. Proteins were eluted from the beads with SDS-PAGE sample buffer, denatured at 37 °C for 2 h, and subjected to immunoblot analysis.

**Flippase assay.** Incorporation of NBD-phospholipids was analyzed by flow cytometry as described[9]. In brief, HeLa cells were detached from dishes in PBS containing 5 mM EDTA, and then collected by centrifugation. Ba/F3 cells were collected from suspension culture by centrifugation. Cells (2 × 10⁵ cells per sample) were washed and equilibrated at 15 °C for 30 min in 100 μl of Hank's balanced salt solution (pH 7.4) containing 1 g/l glucose (HBSS-glucose). An equal volume of 2 μM NBD-phospholipid in HBSS-glucose was added to the cell suspension and incubated at 15 °C. At each time point, 200 μl of the cell suspension was mixed with 200 μl of ice-cold HBSS-glucose containing 5% fatty acid-free BSA (Wako) to extract NBD-lipids incorporated into the exoplasmic leaflet of the plasma membrane, as well as unincorporated lipids. Next, the cells were analyzed with a FACSCalibur (BD Biosciences) to measure fluorescence of NBD-lipids incorporated and translocated into the cytoplasmic leaflet of the plasma membrane. Graphs for NBD-lipid flippase activities are expressed as averages ± SD from at least three independent experiments.

**Data availability.** The data supporting the findings of this study are available within the article and its Supplementary Information files, and are available from the corresponding author upon request.

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

## Acknowledgements

We thank Shigekazu Nagata (Osaka University) for providing cell lines, Takanari Inoue (Johns Hopkins University) for providing Lyn-EGFP-FRB plasmid, Minoru Fukuda (Burnham Institute) for providing anti-TGN46 antibody, and Toshio Kitamura (The University of Tokyo) and Hiroyuki Miyoshi (RIKEN BioResource Center) for kindly providing plasmids. This work was supported by JSPS KAKENHI Grant Numbers JP15H01320, JP16H00764, JP17H03655 (to H.-W.S.), and JP17K08270 (to H.T.); by the Takeda Science Foundation (to H.-W.S.); and by the Research Foundation for Pharmaceutical Sciences (to H.-W.S.). T.N. was supported by a JSPS Research Fellowship for Young Scientists.

## Author contributions

H.T. and H.-W.S.: Conceived and designed the experiments. H.T., M.T., T.N., N.T. and K.T.: Performed the experiments. H.T., M.T. and H.-W.S.: Analyzed the data. H.-W.S.: Prepared the manuscript. All authors discussed results and commented on the manuscript.

## Additional information

**Competing financial interests:** The authors declare no competing financial interests.

