## [Peer Review File · Nature Communications]

Reviewers' comments:

Reviewer #1 (Remarks to the Author):

The study by Takatsu et al. reports on an important regulation pathway for the plasma-membrane localized mammalian lipid flippase ATP11C based on endocytosis triggered by protein kinase C mediated phosphorylation at the C terminal region of the transporter. These process leads to down regulation of the flippase activity at the plasma membrane. They also suggest that Ca²⁺ mediated signaling via activation of G-protein coupled receptors can trigger endocytosis of ATP11C. In short, they claim to have uncovered a new regulation pathway for a mammalian lipid flippase in non-apoptotic cells.

While the signaling dependent endocytosis of ATP11C and the role of the C terminal region of the transporter are well documented, the current study falls short in demonstrating that this process occurs under physiological conditions in mammalian cells. Essentially all data are based on PMA or ionophore treated cells or in cells overexpressing G protein coupled receptors. It is therefore important to demonstrate that at least some cell type under physiological conditions show the same type of regulation. Surprisingly, the authors only test the effect of PMA but not ionophore treatment on other members of the P4-ATPase family (Supplementary Figure S1). Based on the fact that Ca²⁺ is a key signal in activating lipid scrambling at the plasma membrane some efforts should be made to examine the impact of ionophore treatment on other members of the P4-ATPase family.

Further key issues with the manuscript are delineated below:

-The authors claim that increasing cytosolic Ca²⁺ triggers internalization of ATP11C and scrambling at the plasma membrane. However, it is not shown that the ionophore indeed triggers Ca²⁺ increase and extracellular Ca²⁺ is involved. Likewise, lipid scrambling for the cells used in this study is not demonstrated. In Supplementary figure S5 the authors even propose a role for Ca²⁺ release from the ER. Essential controls, e.g. EGTA to chelate extracellular calcium, intracellular Ca²⁺ chelation etc, are missing; the work is incomplete.

-The NBD-PS uptake analysis in Figure 1c is not convincing. While ATP11C transfected HeLa cells display a higher PS uptake in comparison to mock transfected cells, reduction of PS uptake upon PMA treatment is in both cases 30%. Thus, it is not possible to conclude that ATP11C activity is reduced in HeLa cells upon PMA treatment.

- Under results "Flippase activity towards PS decreased by PMA in HeLa cells" the authors state that they investigate whether the PS-flippase activity is affected. The assay used at this stage of the manuscript only monitors the lipid flippase activity at the plasma membrane not the PS-flippase activity of the protein.

-Results for ATP11A in Figure 9 panel (d) and (e) are unclear. Upon PMA treatment a 2-fold increase in ATP11A at the cell surface is observed but NBD-PS remains unchanged. Please explain.

-The labeling of Figure 10a and c is confusing; cells in panel (a) have ATP11c not 5-HT; cells in panel (c) have 5-HT and ATP11C. Panel (c) is labeled twice with 30 min – why?

Reviewer #2 (Remarks to the Author):

This paper presents high quality data to support the conclusion that ATP-dependent translocation of PS at the PM, from the outer to the inner leaflet, is regulated in some cells by the localization of ATP11C. The pool of ATP11C at the PM is down-regulated in response to signaling pathways that trigger its endocytosis.

Although the reported observations are convincing, I have a significant issue with the premise underlying the argument presented by the authors.

The P-type ATPases transport at most 100 phospholipids per second, more usually ~10 per

second. This is related to their ability to turn over ATP because transport is coupled to ATP hydrolysis. In contrast phospholipid scramblases of the TMEM16 family, and also rhodopsin, are capable of moving phospholipids at rates in excess of 10,000 per second. Thus it would seem necessary only to activate a scramblase - with no need to downregulate a flippase - in order to achieve cell surface PS exposure. So the argument that down-regulation of ATP11C is a necessary adjunct to scramblase activation needs to be explained. In this context, the authors must cite contemporary reviews of this field: for example, Montigny et al. (2016) *Biochim. Biophys. Acta* 1861, 767–783 (2016); Pomorski and Menon (2016) *Prog. Lipid Res.* 64, 69–84.

The authors make several references to PLSCR1. This protein was originally identified as a Ca²⁺-activated scramblase by Wiedmer and Sims, but is no longer considered to be relevant to scramblase activity in any system. These statements must be removed or qualified heavily to make it clear that PLSCR1 is not relevant. Reviews that should be cited in this context are: Bevers and Williamson (2016) *Physiol Rev.* 96(2):605-45 and Williamson (2016) *Lipid Insights* 8(Suppl 1):41-4.

The authors refer to Xkr8 family proteins as scramblases - this statement is not correct. The only evidence available so far indicates that Xkr8 proteins are required for PS exposure in apoptotic cells. It remains to be determined whether they are scramblases or provide a supporting function.

Reviewer #3 (Remarks to the Author):

Phosphatidylserine is asymmetrically distributed within cellular membranes. Whereas it is synthesized at the ER, its concentration is highest in the PM, in particular in the PM inner leaflet at the expense of the outer leaflet. This transbilayer asymmetry is maintained by phospholipid flippases in the PM and counteracted by scramblases. Takatsu et al. make an interesting observation that in non-apoptotic cells, the activity of one of the major PS flippases, ATP11C, is regulated by clathrin-mediated endocytosis in a Ca / PKC-dependent manner. This is potentially an important finding, as it connects phospholipid flipping at the PM with membrane trafficking and signaling and shows how PS exposure can be regulated in cells that are not undergoing apoptosis, for example during platelet activation. The authors also carefully map the signals in the C-term tail of ATP11C that are important for its internalization.

Whereas the work is interesting for making connections between processes that were previously not studied together (PS flipping, endocytosis, PKC and Ca signaling), one major problem is that relevant literature is not sufficiently well reviewed so the work is not placed well in a broader context. The most precise mapping of PS distribution in mammalian cells to date has been performed by Parton, Grinstein and colleagues (Fairn 2011). This work is highly relevant and should be appropriately cited. Also striking is the complete absence of any mention of membrane contact sites and lipid transfer proteins, which have in recent years become recognized as an important means of maintaining cellular lipid asymmetry and as important in Ca-signaling. In particular, the authors should carefully consider the recent work by de Camilli and colleagues, both for their findings on PS transport to the PM and connections between Ca signaling and lipid transport in mammalian cells. While this work is for now tangentially related to the work presented in this manuscript, it is clear that understanding how PS content of the PM is regulated under different conditions will require making connections between activity of flippases/scramblases, membrane trafficking AND lipid transport mediated by lipid transfer proteins. The work of T. Graham and colleagues in the yeast model may also be important because they made connections between activity of flippases and vesicle trafficking and hopefully the authors are aware of work in yeast. Finally, the authors should explain the relevance of their model systems, in particular the use of Ba/F3 cells.

There is room in both Introduction and Discussion to place the present work in a more general context with regards to phospholipid (PS) homeostasis. The authors note only in passing (in the

Discussion) an important discrepancy in their findings: that ATP11C knockdown does not result in PS-exposure at the cell surface. This could be due to the activity of other PS-specific flippases, as the authors suggest (but they only studied ATP11C and A), but could also be due to compensation by other processes and pathways, as is often the case for lipids. Also, PKC may not regulate ATP11C activity only through endocytosis (Figure 9), and the increase in ATP11A surface levels upon PMA treatment is intriguing. These discrepancies do not mean that the findings that the authors make on the regulation of ATP11C activity are not interesting, but they show that the regulation of PS asymmetry at the PM is complex.

Another general problem of the paper is that much of the microscopy data is qualitative. Whereas the authors cannot be expected to quantitate all of their data, some experiments should be quantitated to make the results convincing or quantitative statements about the data should be omitted (see specific comments). In the quantifications of cellular data in figures 8, 9 and 10, error bars should be included to illustrate variability between experiments.

Specific comments:

1. The Results section can be organized better. Figure 1c should be moved, or the text describing it should be moved. Figure 9 is difficult to follow (too many panels) and it seems that it would be better to have it before Figure 8. The discrepancies noted in Figure 9 should be discussed in the Discussion.

2. The data on the itinerary that ATP11C takes after being endocytosed is not sufficiently convincing (Figures 2 and 3). Some quantification of colocalization with endocytic markers would be very useful here. In Fig. 3b, staining with EEA1 and TfnR varies between time-points and is very unconvincing.

Finally, the authors present no data here on ATP11C recycling after it is endocytosed; this is speculation. There is no visible change in ATP11C localization at early endosomes over time, and the experiments are not designed to follow recycling, so the discussion of recycling should be omitted here. The data on recycling presented in figure 10, in response to serotonin/histamine treatment, are much more convincing (but the graphs should have error bars).

3. The data that maps the sites in the C-terminus of ATP11C that are required for its recycling is convincing. However, as the starting point was comparison between ATP11A and ATP11C, it would be interesting to know how the sequences of the two C-term tails compare. Are the sites identified in ATP11C absent from A? How about the other P4-ATPases?

4. In Figure 1B, the authors cannot say that the surface levels of ATP11A are increased upon PMA treatment. But the results in Fig. 10 are convincing. It is interesting that the surface levels of ATP11A are increased whereas ATP11C is decreased. Some more discussion on the comparison between the two flippases would be welcome.

Major changes in Figures

Newly added data

Supplementary Figure S1 (A23187 panels)

Supplementary Figure S2

Supplementary Figure S5a panels

Modification of existing Figures

Exchange of Figure 8 and Figure 9

Layout of Figure 8

Error bars added in Figures 8, 9, and 10, and Supplementary Figure S5c

Responses to Reviewer's comments

Reviewer #1 (Remarks to the Author):

The study by Takatsu et al. reports on an important regulation pathway for the plasma-membrane localized mammalian lipid flippase ATP11C based on endocytosis triggered by protein kinase C mediated phosphorylation at the C terminal region of the transporter. These process leads to down regulation of the flippase activity at the plasma membrane. They also suggest that Ca²⁺ mediated signaling via activation of G-protein coupled receptors can trigger endocytosis of ATP11C. In short, they claim to have uncovered a new regulation pathway for a mammalian lipid flippase in non-apoptotic cells.

While the signaling dependent endocytosis of ATP11C and the role of the C terminal region of the transporter are well documented, the current study falls short in demonstrating that this process occurs under physiological conditions in mammalian cells. Essentially all data are based on PMA or ionophore treated cells or in cells overexpressing G protein coupled receptors. It is therefore important to demonstrate that at least some cell type under physiological conditions show the same type of regulation.

> Histamine receptor was not exogenously expressed in HeLa cells in Figure 10a and g. Histamine-treatment stimulates endogenous histamine receptor-mediated signaling in HeLa cells. Therefore, this experiment shows the ATP11C endocytosis under physiological conditions.

Surprisingly, the authors only test the effect of PMA but not ionophore treatment on other members of the P4-ATPase family (Supplementary Figure S1). Based on the fact that Ca^{2+} is a key signal in activating lipid scrambling at the plasma membrane some efforts should be made to examine the impact of ionophore treatment on other members of the P4-ATPase family.

> We have added new data in Supplementary Figure S1. A23187 treatment as well as PMA treatment induces endocytosis of ATP11C not other P4-ATPases including ATP11A.

Further key issues with the manuscript are delineated below:

-The authors claim that increasing cytosolic Ca^{2+} triggers internalization of ATP11C and scrambling at the plasma membrane. However, it is not shown that the ionophore indeed triggers Ca^{2+} increase and extracellular Ca^{2+} is involved. Likewise, lipid scrambling for the cells used in this study is not demonstrated. In Supplementary Figure S5 the authors even propose a role for Ca^{2+} release from the ER. Essential controls, e.g. EGTA to chelate extracellular calcium, intracellular Ca^{2+} chelation etc, are missing; the work is incomplete.

> We thank the reviewer for bringing up the Ca-chelating experiment. We have added new data in Supplementary Figure S5a. ATP11C endocytosis by 5-HT treatment was inhibited by pretreatment with BAPTA-AM. Thus, the endocytosis is triggered by increase of cytosolic Ca^{2+} in response to 5-HT.

Although we did not work with scramblases in the present manuscript, the supplementary Figure S6 (Fig. S5 in the original manuscript) is a model based on our current findings and previous reports by others.

-The NBD-PS uptake analysis in Figure 1c is not convincing. While ATP11C transfected HeLa cells display a higher PS uptake in comparison to mock transfected cells, reduction of PS uptake upon PMA treatment is in both cases 30%. Thus, it is not possible to conclude that ATP11C activity is reduced in HeLa cells upon PMA treatment.

> PS-flipping activity is complex because there are several proteins which can flip PS at the plasma membrane although ATP11C seems to be a major PS-flippase in certain cell types. In the present manuscript, PMA-treatment inhibits exogenous ATP11C-mediated PS-flipping activity, but not exogenous ATP11A-mediated PS-flipping activity in HeLa

cells. Therefore, it is reasonable to conclude that ATP11C-mediated PS-flip activity is reduced upon PMA treatment. We also showed ~30% decrease in HeLa cells (Fig. 1c) and ~50% decrease in Ba/F3 cells (Figs. 5d and 8e). The difference of extent of decrease between cell types could be derived from various contribution and/or compensation of other PS-flippases.

- Under results “Flippase activity towards PS decreased by PMA in HeLa cells” the authors state that they investigate whether the PS-flippase activity is affected. The assay used at this stage of the manuscript only monitors the lipid flippase activity at the plasma membrane not the PS-flippase activity of the protein.

> We agree to the reviewer’s comment. We have changed the sentence (page 4).

-Results for ATP11A in Figure 9 panel (d) and (e) are unclear. Upon PMA treatment a 2-fold increase in ATP11A at the cell surface is observed but NBD-PS remains unchanged. Please explain.

> Because PMA-treatment decreased endogenous ATP11C-mediated PS-flipping activity. Therefore, total PS-flipping activity in ATP11A-expressing cells with PMA-treatment might not increase. We have added a sentence in the discussion (page 13).

-The labeling of Figure 10a and c is confusing; cells in panel (a) have ATP11c not 5-HT; cells in panel (c) have 5-HT and ATP11C. Panel (c) is labeled twice with 30 min – why?

> We have revised the figure legends for better understanding. In panel (c), one is EEA1 staining and the other is Lamp-1 staining.

Reviewer #2 (Remarks to the Author):

This paper presents high quality data to support the conclusion that ATP-dependent translocation of PS at the PM, from the outer to the inner leaflet, is regulated in some cells by the localization of ATP11C. The pool of ATP11C at the PM is down-regulated in response to signaling pathways that trigger its endocytosis.

Although the reported observations are convincing, I have a significant issue with the

premise underlying the argument presented by the authors.

The P-type ATPases transport at most 100 phospholipids per second, more usually ~10 per second. This is related to their ability to turn over ATP because transport is coupled to ATP hydrolysis. In contrast phospholipid scramblases of the TMEM16 family, and also rhodopsin, are capable of moving phospholipids at rates in excess of 10,000 per second. Thus it would seem necessary only to activate a scramblase - with no need to downregulate a flippase - in order to achieve cell surface PS exposure. So the argument that down-regulation of ATP11C is a necessary adjunct to scramblase activation needs to be explained. In this context, the authors must cite contemporary reviews of this field: for example, Montigny et al. (2016) *Biochim. Biophys. Acta* 1861, 767–783 (2016); Pomorski and Menon (2016) *Prog. Lipid Res.* 64, 69–84.

> We appreciate the reviewer's constructive comments. We agree that the transport rate of scramblases is much faster than P4-ATPases *in vitro*. Although the activation of a scramblase would be enough for initial PS-exposure, the exposed PS should be retained at the surface to exhibit biological functions. In fact, constitutive activation of TMEM16F exposes PS at the surface but the cells cannot be phagocytosed by macrophage (Segawa et al., *PNAS*, 2011). Therefore, although inhibition of PS-flippase may not be enough for the initial and acute PS-exposure, it will be critical for holding exposed PS for further biological functions. We have added a sentence describing that 'caspase-resistant ATP11C did not allow PS exposure during apoptosis' in the discussion (pages 14 and 15).

We revised the introduction and discussion, and cited the references which reviewer's suggested (pages 3, 14, and 15).

The authors make several references to PLSCR1. This protein was originally identified as a Ca²⁺-activated scramblase by Wiedmer and Sims, but is no longer considered to be relevant to scramblase activity in any system. These statements must be removed or qualified heavily to make it clear that PLSCR1 is not relevant. Reviews that should be cited in this context are: Bevers and Williamson (2016) *Physiol Rev.* 96(2):605-45 and Williamson (2016) *Lipid Insights* 8(Suppl 1):41-4.

> We thank the reviewer for bringing up the point. We have removed the sentence about PLSCR1, revised the introduction and cited some additional references.

The authors refer to Xkr8 family proteins as scramblases - this statement is not correct. The only evidence available so far indicates that Xkr8 proteins are required for PS exposure in apoptotic cells. It remains to be determined whether they are scramblases or provide a supporting function.

> We agree to the reviewer's comment. We have changed the sentences in the introduction and discussion (pages 3 and 14).

Reviewer #3 (Remarks to the Author):

Phosphatidylserine is asymmetrically distributed within cellular membranes. Whereas it is synthesized at the ER, its concentration is highest in the PM, in particular in the PM inner leaflet at the expense of the outer leaflet. This transbilayer asymmetry is maintained by phospholipid flippases in the PM and counteracted by scramblases. Takatsu et al. make an interesting observation that in non-apoptotic cells, the activity of one of the major PS flippases, ATP11C, is regulated by clathrin-mediated endocytosis in a Ca / PKC-dependent manner. This is potentially an important finding, as it connects phospholipid flipping at the PM with membrane trafficking and signaling and shows how PS exposure can be regulated in cells that are not undergoing apoptosis, for example during platelet activation. The authors also carefully map the signals in the C-term tail of ATP11C that are important for its internalization.

Whereas the work is interesting for making connections between processes that were previously not studied together (PS flipping, endocytosis, PKC and Ca signaling), one major problem is that relevant literature is not sufficiently well reviewed so the work is not placed well in a broader context. The most precise mapping of PS distribution in mammalian cells to date has been performed by Parton, Grinstein and colleagues (Fairn 2011). This work is highly relevant and should be appropriately cited. Also striking is the complete absence of any mention of membrane contact sites and lipid transfer proteins, which have in recent years become recognized as an important means of maintaining cellular lipid asymmetry and as important in Ca-signaling. In particular, the authors should carefully consider the recent work by de Camilli and colleagues, both for their findings on PS transport to the PM and connections between Ca signaling and lipid transport in mammalian cells. While this work is for now tangentially related to the work presented in this manuscript, it is clear that understanding how PS content of the PM is regulated under different conditions will

require making connections between activity of flippases/scramblases, membrane trafficking AND lipid transport mediated by lipid transfer proteins. The work of T. Graham and colleagues in the yeast model may also be important because they made connections between activity of flippases and vesicle trafficking and hopefully the authors are aware of work in yeast. Finally, the authors should explain the relevance of their model systems, in particular the use of Ba/F3 cells.

> We appreciate the reviewer's constructive comments. We have revised the manuscript as follows. First, we have revised the 'Introduction' as reviewer has suggested (page 2). We have mentioned about synthesis, homeostasis, and function of PS in the introduction. And we refer some additional references as reviewer has suggested. But we did not mention about the Ca-signaling and PS-turnover in membrane contact sites. Ca-dependent tethering of extended synaptotagmins to the ER-PM contact sites seems to be required for the diacylglycerol turnover (Saheki et al., NCB, 2016). And thus it is difficult to discuss Ca-dependent PS turnover at the ER-PM contacts in a current stage. Second, concerning to the Ba/F3 cells, we have added a sentence in the 'Results and Discussion' (pages 6 and 13).

There is room in both Introduction and Discussion to place the present work in a more general context with regards to phospholipid (PS) homeostasis. The authors note only in passing (in the Discussion) an important discrepancy in their findings: that ATP11C knockdown does not result in PS-exposure at the cell surface. This could be due to the activity of other PS-specific flippases, as the authors suggest (but they only studied ATP11C and A), but could also be due to compensation by other processes and pathways, as is often the case for lipids. Also, PKC may not regulate ATP11C activity only through endocytosis (Figure 9), and the increase in ATP11A surface levels upon PMA treatment is intriguing. These discrepancies do not mean that the findings that the authors make on the regulation of ATP11C activity are not interesting, but they show that the regulation of PS asymmetry at the PM is complex.

> We have revised the 'Introduction and Discussion' sections (including the discrepancies) (pages 2 and 13). We have revised the introduction with regards to PS synthesis and homeostasis as described above.

Another general problem of the paper is that much of the microscopy data is qualitative. Whereas the authors cannot be expected to quantitate all of their data, some experiments

should be quantitated to make the results convincing or quantitative statements about the data should be omitted (see specific comments). In the quantifications of cellular data in figures 8, 9 and 10, error bars should be included to illustrate variability between experiments.

> We have added the error bars in Figures 8, 9, and 10 and revised the corresponding figure legends.

Specific comments:

1. The Results section can be organized better. Figure 1c should be moved, or the text describing it should be moved. Figure 9 is difficult to follow (too many panels) and it seems that it would be better to have it before Figure 8. The discrepancies noted in Figure 9 should be discussed in the Discussion.

> We have moved the text describing Figure 1c (page 4). We reorganized panels of original Figure 9 (Figure 8 in the revised manuscript) for better understanding and moved before original Figure 8 (Figure 9 in the revised manuscript). We discussed the discrepancies in the original Figure 9 (Figure 8 in the revised manuscript) in the 'Discussion'.

2. The data on the itinerary that ATP11C takes after being endocytosed is not sufficiently convincing (Figures 2 and 3). Some quantification of colocalization with endocytic markers would be very useful here. In Fig. 3b, staining with EEA1 and TfnR varies between time-points and is very unconvincing.

> Because PMA treatment alters the morphology of early endosomes as we mentioned in the manuscript (Aballay et al., JCS, 1999), the endosomes become larger upon PMA-treatment. Moreover, A23187-treatment tends to increase the number of small endosomes in the cell periphery (probably by increasing endocytosis). We have added a sentence describing it (page 5). The key point is that despite of those morphological changes in early endosomes, ATP11C localizes to the early/recycling endosomes (EEA1, VPS35, or TfnR-positive endosomes) but not to late endosomes (Lamp-1-positive endosomes).

We think the quantification of colocalization will not provide more convincing evidence for colocalization and thus we did not add the data. Because several parameters (thresholds, background extraction, ROI determination etc) must be changed in each

sample because of morphological alternations of endosomes and changes of signal intensities upon treatment with PMA and A23187.

Finally, the authors present no data here on ATP11C recycling after it is endocytosed; this is speculation. There is no visible change in ATP11C localization at early endosomes over time, and the experiments are not designed to follow recycling, so the discussion of recycling should be omitted here. The data on recycling presented in figure 10, in response to serotonin/histamine treatment, are much more convincing (but the graphs should have error bars).

> We agree that the description of recycling is better to be moved. We have moved the description of recycling to the text describing Figure 10 (page 11).

3. The data that maps the sites in the C-terminus of ATP11C that are required for its recycling is convincing. However, as the starting point was comparison between ATP11A and ATP11C, it would be interesting to know how the sequences of the two C-term tails compare. Are the sites identified in ATP11C absent from A? How about the other P4-ATPases?

> It has been suggested that N- or C-terminal cytoplasmic region play a regulatory role for P-type ATPase since the middle part is critical for the ATPase activity itself. Among P4-ATPases, N- or C-terminus varies suggesting that the N- or C-terminus is involved in regulation of the enzymes. We have added the sequence comparison between ATP11A and ATP11C in Supplementary Figure S2. Because the C-terminus as well as N-terminus is quite different among P4-ATPases, the comparison among other P4-ATPases will not be informative in the current manuscript. Therefore, we included the comparison between ATP11A and ATP11C.

4. In Figure 1B, the authors cannot say that the surface levels of ATP11A are increased upon PMA treatment. But the results in Fig. 10 are convincing. It is interesting that the surface levels of ATP11A are increased whereas ATP11C is decreased. Some more discussion on the comparison between the two flippases would be welcome.

> We had mentioned that 'ATP11A level slightly increased' in the original text but we did not discuss it. We have mentioned about it in the discussion of the revised manuscript.

REVIEWERS' COMMENTS:

Reviewer #1 (Remarks to the Author):

The authors have submitted a substantially improved revision of the manuscript. In particular, the authors now provided new data sets on showing that the PMA effect is specific for ATP11C and intracellular calcium increase is required for internalization of ATP11C. Furthermore, the authors corrected the labeling of some figures. However, the model proposed in supplemental figure S6 should be more strengthened by demonstrating lipid scrambling in the established HeLa cells stably co-expressing Gq-coupled serotonin receptor (5-hydroxytryptamine (HT) receptor 2A; 5-HT2A-R) and ATP11C-HA after serotonin treatment by using an established, commercially available Annexin-V binding assay. This assay is straight forward and well established.

Reviewer #2 (Remarks to the Author):

The authors have addressed my comments in a general way. The specific issue that was most important in my mind related to the relative rates of ATPase-mediated flipping versus scrambling. These rates differ by at least 2 orders of magnitude. On line 500 of the paper the authors raise this point briefly and cite data to the effect that ATPases must be downregulated even as scramblases are turned on. This is fine. However, it would still be good to see a few lines of text in the same context - this could be inserted as a new sentence on line 497, before the final sentence beginning with 'During ...', and here the authors could say something about rates and cite ref 69 for the details.

Reviewer #3 (Remarks to the Author):

The authors have largely addressed my concerns. Their manuscript contains a great amount of work.

However, the issue of the di-leucine signal in the C-term tail of ATP11C could be clarified more. Using chimeras between ATP11A and ATP11C, the authors have shown that the C-term tail of ATP11C is required for the internalization of the PS-flippase. They also show that an [ED]xxxL[LI] type signal in the tail is required for internalization. The obvious question is, is this signal making the difference between ATP11C and 11A. The authors now provide an alignment of the two C-term sequences in Suppl. Fig. S2. The di-leucine repeat should be highlighted in this figure. I note that 11C contains the sequence SVRPLL, and in 11A the sequence reads STIFML. Both S1116 and L1121 are therefore conserved between the two proteins; is the L1120 to M substitution preventing internalization of ATP11A? Given the amount of data in this manuscript, it seems unfair to ask the authors to perform this experiment at this point, but they should at least address this question more carefully in the text. The first L in the di-leucine signal is generally supposed to be the invariant one, whereas the second one can also be an I; but looking through the literature, the issue is not so clear. For example, Braulke and Bonifacino (2008) (a more recent review than the one that the authors cite), lists an AP-2 interacting signal 'NEQLPML' (see table Table 1). The signal in ATP11A might therefore work as well. It is of course possible that something else in the ATP11A C-term is preventing internalization, but, again, it should be noted in the text that this issue is not so clear. Related to this, it would be better to replace the reference 44 with the reference Kelly et al., Nature, 2008 (also from the Owen lab), as in this paper the authors look directly at how different di-leucine signals fit into the AP-2 binding site (see in particular Table S2, where the authors measure directly the binding affinities of some phosphorylated di-L peptides).

Responses to Reviewer's comments

Reviewer #1 (Remarks to the Author):

The authors have submitted a substantially improved revision of the manuscript. In particular, the authors now provided new data sets on showing that the PMA effect is specific for ATP11C and intracellular calcium increase is required for internalization of ATP11C. Furthermore, the authors corrected the labeling of some figures. However, the model proposed in supplemental figure S6 should be more strengthened by demonstrating lipid scrambling in the established HeLa cells stably co-expressing Gq-coupled serotonin receptor (5-hydroxytryptamine (HT) receptor 2A; 5-HT_{2A}-R) and ATP11C-HA after serotonin treatment by using an established, commercially available Annexin-V binding assay. This assay is straight forward and well established.

> We previously had tried the experiments but we could not detect the exposure of PS using Annexin V assay. As reviewer mentioned, Annexin-V binding assay is well established for global PS exposure such as during apoptosis and platelet activation. But it does not seem to work in transient and local PS exposure. We may need another experimental system to detect such transient and local PS exposure. Supplemental figure S6 is a current model. Therefore, we had described 'local asymmetrical PS distribution' in the Discussion (line 435 in the previous revised-manuscript).

Reviewer #2 (Remarks to the Author):

The authors have addressed my comments in a general way. The specific issue that was most important in my mind related to the relative rates of ATPase-mediated flipping versus scrambling. These rates differ by at least 2 orders of magnitude. On line 500 of the paper the authors raise this point briefly and cite data to the effect that ATPases must be downregulated even as scramblases are turned on. This is fine. However, it would still be good to see a few lines of text in the same context - this could be inserted as a new sentence on line 497, before the final sentence beginning with 'During ...', and here the authors could say something about rates and cite ref 69 for the details.

> We have added following sentence as reviewer mentioned.

(Although the activation of scramblases might be enough for initial PS exposure, stable PS exposure for further cellular functions and/or PS-selectivity would be attributed to

the inhibition of ATP11C since scramblases do not seem to have a substrate specificity.)

Reviewer #3 (Remarks to the Author):

The authors have largely addressed my concerns. Their manuscript contains a great amount of work.

However, the issue of the di-leucine signal in the C-term tail of ATP11C could be clarified more. Using chimeras between ATP11A and ATP11C, the authors have shown that the C-term tail of ATP11C is required for the internalization of the PS-flippase. They also show that an [ED]xxxL[LI] type signal in the tail is required for internalization. The obvious question is, is this signal making the difference between ATP11C and 11A. The authors now provide an alignment of the two C-term sequences in Suppl. Fig. S2. The di-leucine repeat should be highlighted in this figure. I note that 11C contains the sequence SVRPLL, and in 11A the sequence reads STIFML. Both S1116 and L1121 are therefore conserved between the two proteins; is the L1120 to M substitution preventing internalization of ATP11A? Given the amount of data in this manuscript, it seems unfair to ask the authors to perform this experiment at this point, but they should at least address this question more carefully in the text.

The first L in the di-leucine signal is generally supposed to be the invariant one, whereas the second one can also be an I; but looking through the literature, the issue is not so clear. For example, Braulke and Bonifacino (2008) (a more recent review than the one that the authors cite), lists an AP-2 interacting signal 'NEQLPML' (see table Table 1). The signal in ATP11A might therefore work as well. It is of course possible that something else in the ATP11A C-term is preventing internalization, but, again, it should be noted in the text that this issue is not so clear. Related to this, it would be better to replace the reference 44 with the reference Kelly et al., Nature, 2008 (also from the Owen lab), as in this paper the authors look directly at how different di-leucine signals fit into the AP-2 binding site (see in particular Table S2, where the authors measure directly the binding affinities of some phosphorylated di-L peptides).

> As reviewer mentioned, STIFML sequences in ATP11A seem to be similar to SVRPLL in ATP11C. However, ATP11A did not endocytose by Ca-mediated PKC activation. We have added a sentence to describe it in Discussion (Although ATP11A also contains a STIFML sequence, which is similar to dileucine-based signals, in its C-terminal cytoplasmic region (Supplementary Fig. 2), it could not serve as the

endocytic signal at least by Ca^{2+} -mediated PKC activation.) and highlighted the sequence in Supplementary Figure 2.

Even if the sequences (STIFML) in ATP11A might be another regulatable trafficking motif, it does not work at least by Ca-mediated PKC activation. It might be regulated by other mechanisms. Therefore, the substitution of L to M will not be necessary for the current ATP11C story. It would be a future work.